EMBO
Molecular Medicine

# Co-targeting BET and MEK as salvage therapy for MAPK and checkpoint inhibitor-resistant melanoma

Ileabett M Echevarría-Vargas[1], Patricia I Reyes-Uribe[1], Adam N Guterres[1], Xiangfan Yin[1], Andrew V Kossenkov[1], Qin Liu[1], Gao Zhang[1], Clemens Krepler[1], Chaoran Cheng[2], Zhi Wei[2], Rajasekharan Somasundaram[1], Giorgos Karakousis[3,4], Wei Xu[3], Jennifer JD Morrissette[5,6], Yiling Lu[7], Gordon B Mills[7], Ryan J Sullivan[8], Miao Benchun[8], Dennie T Frederick[8], Genevieve Boland[9], Keith T Flaherty[8], Ashani T Weeraratna[10,11], Meenhard Herlyn[1,10], Ravi Amaravadi[4,12], Lynn M Schuchter[4,12], Christin E Burd[13], Andrew E Aplin[14], Xiaowei Xu[4,7] & Jessie Villanueva[1,10,*]

## Abstract

Despite novel therapies for melanoma, drug resistance remains a significant hurdle to achieving optimal responses. NRAS-mutant melanoma is an archetype of therapeutic challenges in the field, which we used to test drug combinations to avert drug resistance. We show that BET proteins are overexpressed in NRAS-mutant melanoma and that high levels of the BET family member BRD4 are associated with poor patient survival. Combining BET and MEK inhibitors synergistically curbed the growth of *NRAS*-mutant melanoma and prolonged the survival of mice bearing tumors refractory to MAPK inhibitors and immunotherapy. Transcriptomic and proteomic analysis revealed that combining BET and MEK inhibitors mitigates a MAPK and checkpoint inhibitor resistance transcriptional signature, downregulates the transcription factor TCF19, and induces apoptosis. Our studies demonstrate that co-targeting MEK and BET can offset therapy resistance, offering a salvage strategy for melanomas with no other therapeutic options, and possibly other treatment-resistant tumor types.

**Keywords** BET; combination therapy; drug resistance; melanoma; mutant NRAS

**Subject Categories** Cancer; Pharmacology & Drug Discovery; Skin

See also: **Y Yu** (May 2018)

## Introduction

Promising new therapies have emerged for *BRAF*-mutant melanoma patients, but *NRAS*-mutant (*NRAS*[Mut]) melanoma, like other *RAS*-driven tumors, continues to have poor prognosis and limited therapeutic options (Sullivan & Flaherty, 2013; Johnson *et al*, 2014; Posch *et al*, 2016; Vu & Aplin, 2016; Wong & Ribas, 2016). Somatic mutations in *NRAS* account for approximately 26% of all malignant melanoma (Hodis *et al*, 2012). Additionally, development of secondary *NRAS* mutations is a frequent mechanism for acquired resistance to BRAF inhibitors (Nazarian *et al*, 2010; Van Allen *et al*, 2014). Currently, there are few effective therapeutic options for *NRAS*-driven melanoma (Johnson & Puzanov, 2015). Clinical studies have evaluated compounds that target RAS effectors, mainly inhibitors of the mitogen-activated protein kinase (MAPK, e.g., MEK inhibitors) and phosphatidylinositol-3 kinase (PI3K) signaling pathways (Kwong & Davies, 2014). However, the therapeutic efficacy of these drugs as single agents is modest

1   Molecular & Cellular Oncogenesis Program, The Wistar Institute, Philadelphia, PA, USA
2   College of Computing Sciences, New Jersey Institute of Technology, Newark, NJ, USA
3   Abramson Cancer Center, University of Pennsylvania, Philadelphia, PA, USA
4   Department of Surgery, Hospital of the University of Pennsylvania, Philadelphia, PA, USA
5   Center for Personalized Diagnostics, Hospital of the University of Pennsylvania, University of Pennsylvania, Philadelphia, PA, USA
6   Department of Pathology and Laboratory Medicine, Hospital of the University of Pennsylvania, Philadelphia, PA, USA
7   Department of Systems Biology, The University of Texas MD Anderson Cancer Center, Houston, TX, USA
8   Massachusetts General Hospital Cancer Center, Harvard Medical School, Boston, MA, USA
9   Department of Surgery, Massachusetts General Hospital, Harvard Medical School, Boston, MA, USA
10  Melanoma Research Center, The Wistar Institute, Philadelphia, PA, USA
11  Immunology, Microenvironment and Metastasis Program, The Wistar Institute, Philadelphia, PA, USA
12  Department of Medicine, Hospital of the University of Pennsylvania, Philadelphia, PA, USA
13  Departments of Molecular Genetics and Cancer Biology and Genetics, Ohio State University, Columbus, OH, USA
14  Department of Cancer Biology and Sidney Kimmel Cancer Center, Thomas Jefferson University, Philadelphia, PA, USA
    *Corresponding author. Tel: +1 215 495 6818; E-mail: jvillanueva@wistar.org

(Johnson & Puzanov, 2015). Here, we focused on identifying novel approaches targeting "non-oncogene addictions", which have the potential to induce cell death of $NRAS^{Mut}$ melanoma when combined with inhibitors of RAS effectors, and explored the efficacy of this strategy in targeted and immune checkpoint inhibitor-resistant melanoma.

Melanoma, like other cancers, is driven by genetic and epigenetic alterations. Epigenetic mechanisms implicated in melanoma-genesis include altered gene expression via promoter hypo- or hypermethylation, histone modification, chromatin remodeling, and expression of non-coding RNAs (Sarkar et al, 2015). For example, hypermethylation of the CDKN2A tumor suppressor promoter occurs in ~ 20% of primary melanomas and is associated with reduced patient survival (Straume et al, 2002). Oncogenic pathways, like mutant NRAS, can modulate and interact with the cancer epigenome; hence, epigenetic factors could constitute therapeutic targets for $NRAS^{Mut}$ tumors (Besaratinia & Tommasi, 2014). For example, RAS/MAPK signaling promotes expression of the chromatin remodeler EZH2, which mediates chromatin compaction via histone H3K27 methylation, thereby repressing expression of its target genes (Fujii et al, 2011; Hou et al, 2012). Primary and metastatic melanomas express aberrantly high levels of EZH2, which is associated with poor survival (Zingg et al, 2015). Other epigenetic regulators that are commonly deregulated in melanoma include ATP-dependent chromatin remodelers belonging to the SWI/SNF family, mediators of DNA de/methylation such as TET2 and IDH1/2, and covalent modifiers of histones, including histone deacetylases (HDAC9) and methyltransferases (SETD2) (Hayward et al, 2017). One group of epigenetic regulators that has emerged as promising therapeutic targets for cancer is the Bromodomain and Extra-terminal Domain (BET) family of proteins (Filippakopoulos & Knapp, 2014; Ugurel et al, 2016). Bromodomains are known to bind acetylated lysine residues in the N-terminal tail of histones and non-histone proteins, serving as scaffolds facilitating gene transcription and regulating many cellular processes including DNA replication and cell cycle progression (Shi & Vakoc, 2014). Several small molecule inhibitors of BET proteins (BETi) have been developed as potential anti-cancer agents, and some are currently undergoing clinical investigation in various tumor types including melanoma (NCT02259114, NCT02369029, NCT01987362, NCT02683395, NCT01587703; Brand et al, 2015). In this study, we evaluated the efficacy of BETi when combined with MEKi in restraining $NRAS^{Mut}$ melanoma and offsetting drug resistance. Our data support the premise that there is a unique synergistic vulnerability exposed by combining BET and MEK inhibitors, and that this combination could be used as a salvage strategy for targeted- and immune checkpoint inhibitor-resistant melanoma.

## Results

### BRD4 as a molecular target for NRAS-mutant melanoma

To identify therapeutic vulnerabilities in NRAS-mutant melanoma, we explored different potential targets for expression in the TCGA skin cutaneous melanoma dataset (SKCM, Provisional 2017; www.cbioportal.org) (Cerami et al, 2012; Gao et al, 2013).

This analysis revealed that high BRD4 mRNA expression was associated with poor patient survival ($P = 0.001$; Fig 1A) and disease-free survival ($P = 0.0008$; Fig EV1) in NRAS-mutant melanoma patients, but not in other genetic cohorts (Appendix Fig S1). We next performed immunohistochemical analysis of biopsies from 54 patients with genetically diverse metastatic melanoma and confirmed high expression of BRD4 in $NRAS^{Mut}$ tumors; BRD4 levels were markedly higher than in tumors harboring mutant BRAF or wild-type for BRAF and NRAS (WT) (Fig 1B and C). To determine the effect of BRD4 blockade, we silenced BRD4 in NRAS-mutant melanoma cells (Appendix Fig S2). Depletion of BRD4 decreased the viability of NRAS-mutant melanoma cells (Fig 1D), but induced only modest apoptosis (Fig 1E). These data suggest that BRD4 plays an important role in NRAS-mutant melanoma and it is necessary for proliferation of these cells.

We next evaluated the effect of JQ-1 (a prototype BET inhibitor) on cell viability and determined the half-maximal inhibitory concentration ($IC_{50}$) of JQ-1 in NRAS-mutant melanoma cells intrinsically resistant to MAPK inhibitors as well as non-transformed cells (Fig 2A and Appendix Table S1). JQ-1 decreased the viability of NRAS-mutant melanoma cells; moreover, sensitivity to JQ-1 inversely correlated with BRD4 protein levels (Pearson's correlation coefficient = $-0.759$, $P = 0.018$; Figs 2A–C and EV2) but not with BRD2 or BRD3 levels (Fig 2D–H).

To identify more effective therapies for NRAS-mutant melanoma, we evaluated combinations of JQ-1 with inhibitors of RAS effectors that are undergoing clinical evaluation for NRAS-mutant melanoma patients. We selected inhibitors of MEK (PD0325901; PD901), CDK4/6 (PD0332991; PD991), and PI3K (BKM120) (Fig EV2). We found that the MEK inhibitor (MEKi) PD901 potently synergized with JQ-1 (Figs EV2 and EV3A). Treatment of NRAS-mutant melanoma cells with a single dose of JQ-1 (0.5 μM) in combination with PD901 (0.1 μM) substantially impaired colony formation compared to single agent treatment (Fig EV3B). Single doses of either compound as a single agent transiently restrained cell proliferation, but this effect was not sustained at 14 days (Fig EV3C, top panels). In contrast, a single dose of the combination treatment led to sustained inhibition of melanoma cell proliferation (Fig EV3C, bottom panels), whereas it only transiently inhibited the growth of non-transformed cells (Fig EV3C). While BET or MEK inhibitors predominantly induced cytostatic effects as single agents, the combination of both compounds triggered significant apoptosis selectively in NRAS-mutant melanoma cells without affecting non-transformed cells (Fig EV3D). We further explored the efficacy of this combination using the structurally-related BET family inhibitor OTX-015 (MK-8628), which is currently in phase II clinical trials for solid tumors (NCT02698176; NCT02296476; Odore et al, 2016; Amorim et al, 2016) and the FDA-approved MEKi trametinib. Combining trametinib with the clinically relevant BETi OTX-015 more potently induced cell death compared to single agents (Appendix Fig S3), further supporting the notion that the combination of BET and MEK inhibitors elicit cytotoxic effects in NRAS-mutant melanoma.

In similar experiments, we found that although combining JQ-1 with the CDK4/6 inhibitor, PD991, appeared to be effective in NRAS-mutant melanoma (Fig EV2, and Appendix Fig S4A and B), this combination significantly inhibited the proliferation of

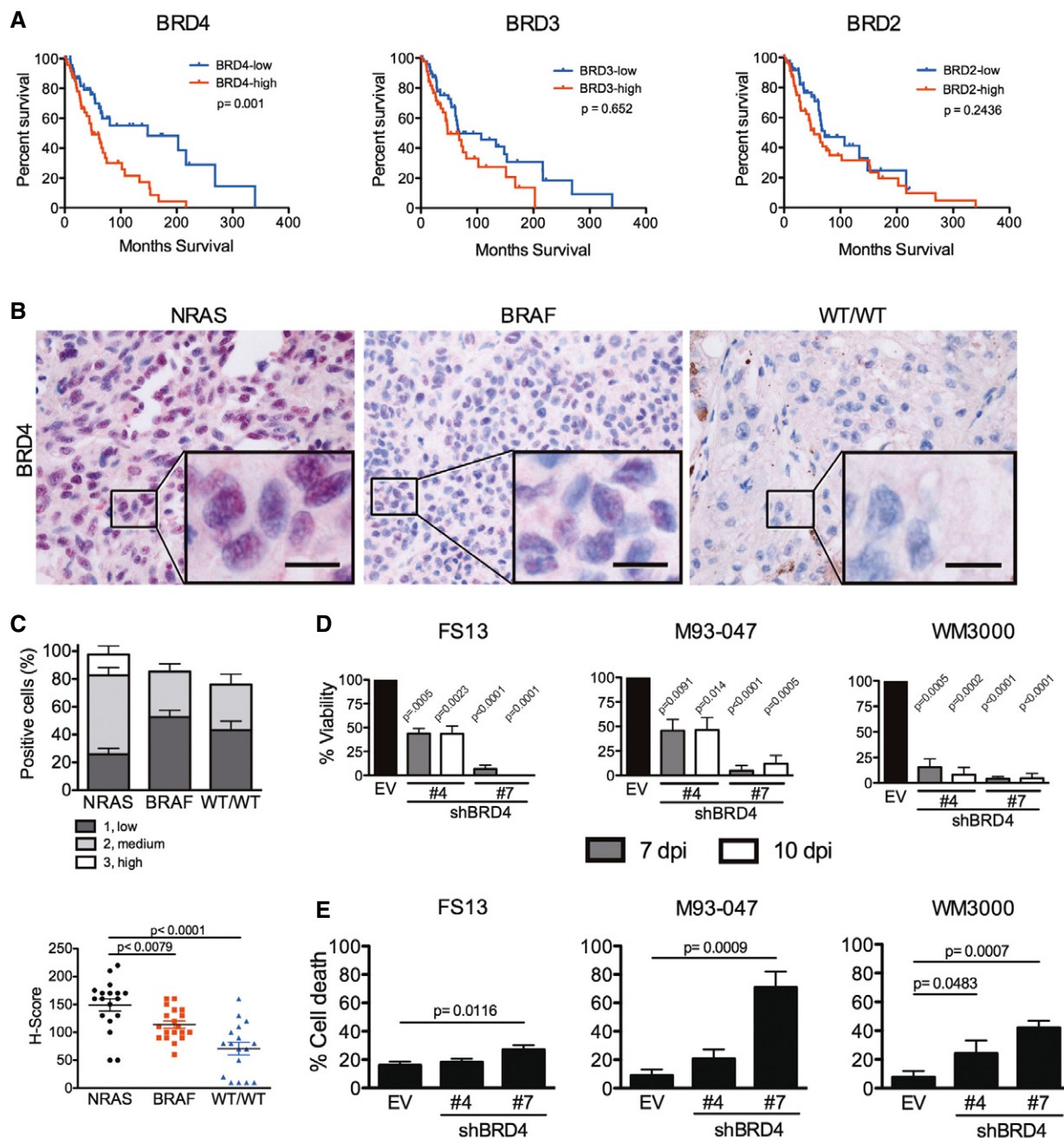

**Figure 1.  BRD4 is associated with poor patient survival and constitutes a promising target for NRAS[Mut] melanoma.**

A     NRAS-mutant melanoma samples (*n* = 98) were analyzed from the skin cutaneous melanoma TCGA database. Samples were classified into high or low BRD4, BRD3, and BRD2 expressing groups according to the median tissue mRNA expression levels. Overall survival Kaplan–Meier curves for BRD4, BRD3, BRD2 in the NRAS-mutant group are shown; *P*-values were calculated by long-rank tests comparing the two Kaplan–Meier curves.

B, C   Fifty-four patient samples were categorized in subgroups based on mutation status: *NRAS* (*n* = 18), *BRAF* (*n* = 19), or *BRAF-WT/NRAS-WT* (WT/WT; *n* = 17). (B) BRD4 expression was assessed by immunohistochemistry (IHC) in biopsies from patients with metastatic melanoma. BRD4 staining is localized in the nucleus. High magnification (40×) representative images are shown; the scale bar represents 20 μm. (C) BRD4 expression by IHC was scored blindly as low (1), medium (2), or high (3) for each sample. Scatter plot showing BRD4 expression in each subgroup ranging from total absence of BRD4 in the tumor (*H*-score = 0) to high BRD4 expression (*H*-score = 300). Each point represents the *H*-score from a single tumor sample. The horizontal line represents the mean *H*-score ± SEM.

D     BRD4 was silenced using two different hairpins (#4 and #7) in NRAS-mutant melanoma cells (FS13, M93-047, and WM3000). Cell viability was determined by MTT assays 7 or 10 days post-transduction (dpi) and calculated relative to empty vector (EV) control.

E     Cell death (Annexin V[+]/PI[+]) was analyzed by flow cytometry 7 dpi.

Data information: For (D, E), data represent the mean of three independent experiments ± SEM. *P*-values when comparing each condition with its corresponding control (empty/non-targeting vector) was calculated using Student's *t*-test are indicated in each figure.

Source data are available online for this figure.

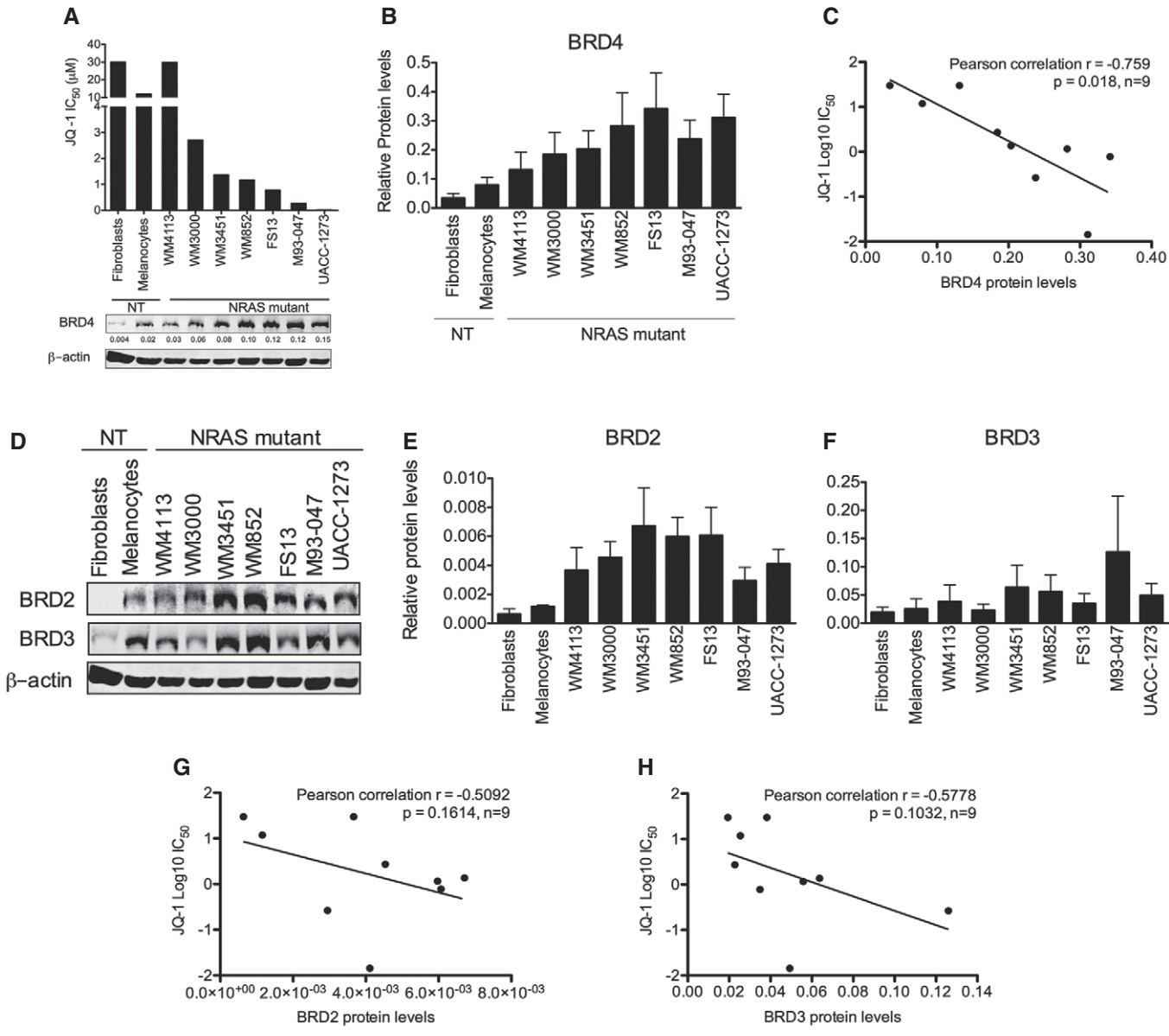

**Figure 2. BRD4 expression levels are associated with sensitivity to BET inhibition in NRAS-mutant melanoma.**

A    Cells were treated with increasing doses of JQ-1 for 3 days, and the number of viable cells was determined by MTT assays. Concentrations inhibiting 50% of cell growth ($IC_{50}$) were calculated at day 3 using GraphPad Prism V5.0a. Bottom panel: Expression of BRD4 was assessed by immunoblotting; β-actin was used as loading control. Representative Western blot is shown.

B    Quantification of BRD4 protein levels from immunoblots is shown. Membranes were scanned and quantified using the LI-COR Odyssey system; average protein levels from three independent experiments ± SEM are depicted in the bar graphs.

C    Linear correlation between BRD4 protein levels and JQ-1 $IC_{50}$ was assessed using Pearson's correlation coefficient. Data analysis was performed using Stata version 13.

D    Expression of BRD2 and BRD3 was evaluated by Western blot in a panel of NRAS-mutant melanoma and non-transformed cell lines.

E, F    Quantification of BRD2 (E) and BRD3 (F) protein levels from immunoblots is shown. Membranes were scanned and quantified using the LI-COR Odyssey system; average protein levels from three independent experiments ± SEM are depicted in the bar graphs.

G, H    Linear correlation between BRD2 (G) and BRD3 (H) protein levels and JQ-1 $IC_{50}$ was assessed using Pearson's correlation coefficient. Data analysis was performed using Stata version 13.

Source data are available online for this figure.

non-transformed cells (Appendix Fig S4B). Additionally, for the PI3K inhibitor, BKM120, concentrations above 1 μM were needed to potentiate the effect of JQ-1 (Fig EV2, and Appendix Fig S4A and C).

We reasoned that these doses of BKM120 might not be achievable *in vivo*. Therefore, we opted to further pursue the combination of BET and MEK inhibitors.

## Co-targeting BET and MEK inhibits the growth of NRAS-mutant melanoma and increases survival of tumor-bearing mice

We next evaluated the efficacy of BET and MEK inhibitors in 3D collagen-embedded NRAS-mutant melanoma spheroids, as these more closely resemble the behavior of human melanoma *in vivo* (Smalley *et al*, 2007, 2008). Combined treatment with JQ1 or OTX-015 plus PD901 triggered substantial cell death of NRAS-mutant melanoma spheroids, compared to vehicle or single agent treatments (Fig 3A and B). We next assessed the efficacy of BETi/MEKi combinations *in vivo*. Treatment of M93-047 NRAS-mutant xenografts with vehicle, BETi (JQ-1 or OTX-015), MEKi (PD901), or BETi plus MEKi, demonstrated that BETi/MEKi combinations led to sustained tumor growth inhibition compared to single agents (Fig 3C and D, and Appendix Table S2). Of note, whereas BETi alone did not prevent tumor growth (Fig 3C and D), the MEKi PD901 decreased tumor growth rate, but this effect was not sustained and tumors resumed growth after 2 weeks of treatment. Moreover, the BETi/MEKi combination significantly increased the survival of mice that were treated for 21 days, without evidence of toxicity (Fig 3E and F, and Appendix Fig S5A and B).

To better understand the mechanism whereby MEK and BET inhibitors cooperate to inhibit the growth of NRAS-mutant melanoma, we performed RNA-sequencing and proteomic analysis of cells treated with single-agent JQ-1, PD901, or the combination of both drugs. Differential expression analysis showed that the combination of JQ-1 and PD901 affected 7,909 genes (FDR < 5%) compared to vehicle-treated cells (Fig 4A). The BETi/MEKi combination differently affected 2,557 genes (FDR < 5%) compared to JQ-1, and 5,541 genes (FDR < 5%) compared to PD901. Overlap of the three comparisons identified 1,129 genes significantly more affected by the combination treatment than by JQ-1 or PD901 alone (Table EV1). Ingenuity Pathway Analysis (IPA) of these 1,129 genes revealed significant inhibition ($P < 10^{-4}$, inhibition Z-score < −2) of functions leading to cell growth, including DNA synthesis ($n = 51$ genes), survival ($n = 153$ genes), metastasis ($n = 102$ genes), migration ($n = 213$ genes), M-phase ($n = 56$ genes), and proliferation ($n = 386$ genes). Conversely, pathways leading to death ($n = 551$ genes) were significantly activated ($P < 10^{-4}$, activation Z-score < 2; Fig 4B). Genes that were synergistically downregulated in response to JQ1/PD901 combination treatment were independently validated by qRT–PCR in two NRAS-mutant patient-derived cell lines (Fig 4C, Appendix Fig S6A and B, and Appendix Table S3), including genes that regulate cell division, DNA replication, and apoptosis (Fig 4C, and Appendix Fig S6A and B). Notably, three transcriptional regulators were significantly repressed by combination treatment compared to single agents: TCF19, E2F1, and E2F3 (Fig 4C and Appendix Fig S6C). Consistently, depletion of BRD4 partially downregulated TCF19; this effect was substantially enhanced by treatment with MEKi. These results suggest that BET family proteins and MEK jointly regulate the expression of TCF19 (Appendix Fig S6D).

We next performed reverse phase protein analysis (RPPA) to explore the effect of co-targeting BET and MEK on signaling networks. This analysis revealed that the combination of BETi/MEKi induced downregulation of PLK1, inhibition of phosphorylation of the retinoblastoma protein (pRb), and upregulation of the cdk inhibitor p27 (Fig 4D). RPPA analysis also revealed that

co-targeting BET and MEK upregulates the pro-apoptotic factor Bim and triggers caspase-7 cleavage (> 1.5-fold; Fig 4D). We selected candidates that were found to be markedly and consistently affected by the combination treatment at the mRNA and protein level in all the cell lines evaluated by the two different BET inhibitors for further validation by immunoblotting in a panel of NRAS-mutant cell lines (Fig 4E and Appendix Fig S6E). Activation of the effector caspase-7 was observed in all NRAS-mutant melanoma cell lines following co-inhibition of BET and MEK.

Altogether, these results raised the possibility that downregulation of TCF19 and E2F1/3-dependent targets (e.g., CDC25C, CCNA2, cyclin E2, BIRC5) and increased BIM levels could contribute to the activation of apoptotic pathways triggered by the combination of BETi/MEKi (Alla *et al*, 2010). We noted that TCF19 is expressed in NRAS-mutant melanoma cells and in melanocytes, albeit at lower levels (Fig 5A). Additionally, TCF19 levels positively correlated with BRD4 protein levels and sensitivity to BETi in NRAS-mutant melanoma cells ($r = -0.740$, $P = 0.023$; Figs 2A and B, and 5A and B). We next explored the biological effects of blocking TCF19 in NRAS-mutant cells, as the role of this PHD-type zinc finger-containing domain transcription factor has not been previously investigated in melanoma. Depletion of TCF19 with two different shRNA constructs led to an increase of cells in G2/M followed by apoptosis of NRAS-mutant melanoma cells (Fig 5C–E and Appendix Fig S7). This increase in cell death induced by a shRNA targeting the untranslated region (3′ UTR) of TCF19 was attenuated by co-transduction with a TCF19 open reading frame (ORF). The requirement of TCF19 for melanoma cell survival, suggests that this transcription factor could play an important role in this disease. Indeed, analysis of the TCGA cutaneous melanoma dataset revealed that BRD4 and TCF19 were positively associated in melanoma and that the levels of TCF19 inversely correlated with patient's survival ($P = 8.3 \times 10^{-3}$; Fig 5F and Appendix Fig S8). However, there was no statistically significant correlation with a particular genetic cohort. Taken together, these results indicate that the combination of BETi/MEKi perturbs the cell cycle machinery and activates apoptotic signaling in NRAS-mutant melanoma cells in part by synergistically downregulating the PHD-type zinc finger domain containing transcription factor TCF19.

## Concurrent BET and MEK inhibition downregulates TCF19 and overcomes resistance to targeted and immunotherapy

While immunotherapy has shown remarkable efficacy in melanoma and other cancers, not all patients respond (Luke *et al*, 2017). Likewise, vertical inhibition of MAPK signaling by combining BRAF and MEK inhibitors is now approved for BRAF-mutant melanoma, but responses are transient and resistance to this combination is a pressing issue (Moriceau *et al*, 2015; Welsh *et al*, 2016). Mutations in NRAS confer both intrinsic and acquired resistance to BRAF and MEK inhibitors (Poulikakos *et al*, 2011; Boussemart *et al*, 2016; Fiskus & Mitsiades, 2016). We noted that BETi/MEKi combination synergistically downregulated a transcriptional signature associated with resistance to both MAPK and checkpoint inhibitors suggestive of an epigenetic-mediated mechanism (Hugo *et al*, 2016; Fig EV4). We therefore reasoned that BETi/MEKi could be a strategy to overcome resistance to targeted- and immune-therapy. We first tested the efficacy of combining BET and MEK inhibitors in a

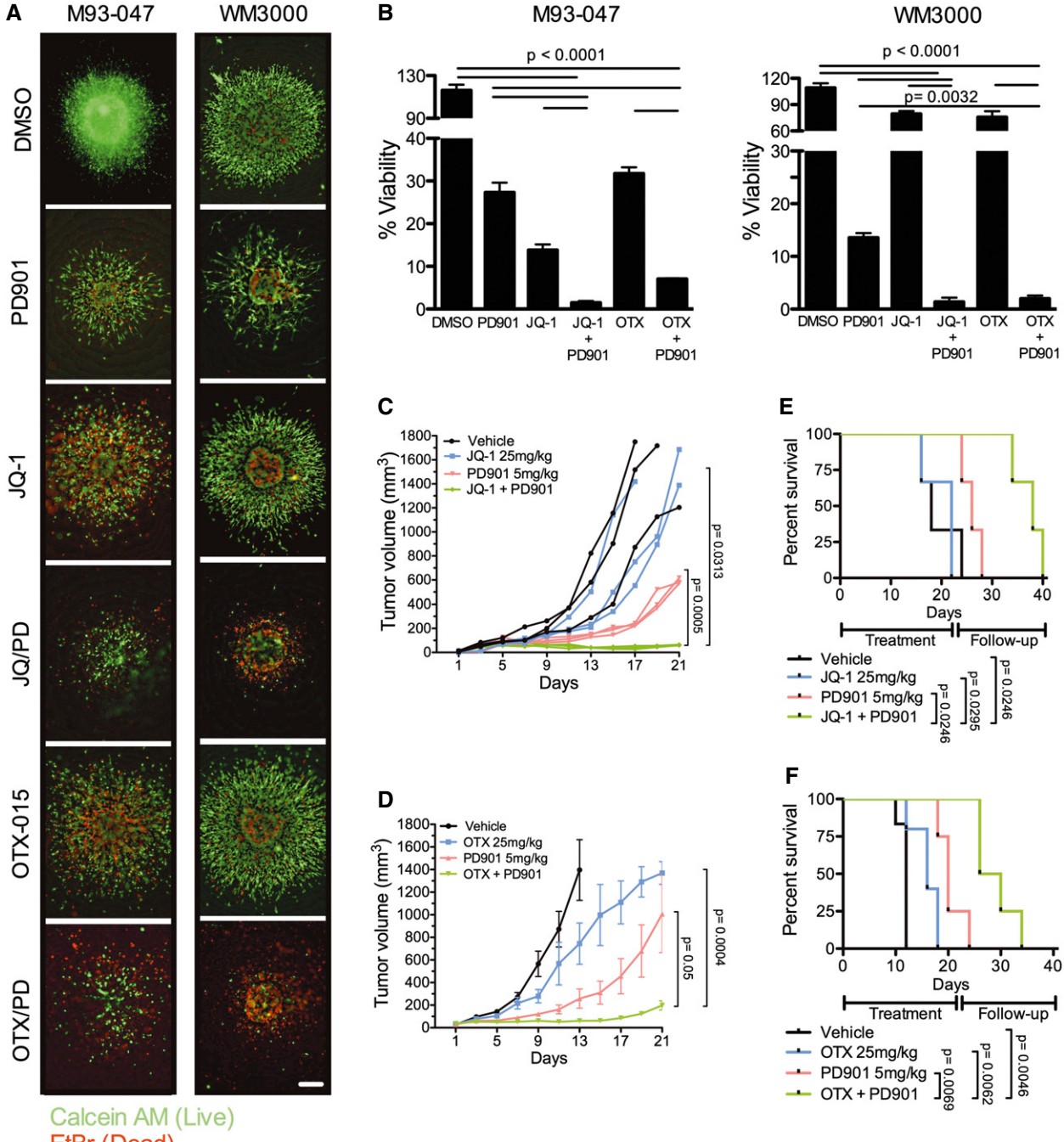

Calcein AM (Live)
EtBr (Dead)

**Figure 3. BETi potentiate the efficacy of MEKi in NRAS-mutant melanoma.**

A NRAS-mutant melanoma cells grown as collagen-embedded 3D spheroids were treated with DMSO, 0.1 μM MEKi PD901, 0.5 μM BETi (JQ-1 or OTX-015), or BETi + MEKi combination for 5 days. Spheroids were stained with Calcein (AM) (green; live cells) and EtBr (red; dead cells) and imaged using a fluorescence microscope. Representative images (4×) of three replicates are shown; the scale bar represents 250 μm.

B Spheroid viability was determined by Alamar Blue assay after 5 days of treatment. Percent cell viability (%) was calculated relative to DMSO-treated spheroids. Data represent the average of three independent experiments ± SEM. *P*-values were calculated using Student's *t*-test.

C–F Five-week-old NSG mice were injected subcutaneously with $1 \times 10^{6}$ M93-047 cells. Mice were randomized into four treatment groups: vehicle control, BETi [JQ-1 (25 mg/kg ip.qd × 21 days) or OTX-015 (25 mg/kg po.qd × 21 days)], PD901 (5 mg/kg po.qd × 21 days), or combination (C: JQ-1 + PD901; *n* = 3 mice/group) (D: OTX-015 + PD901; *n* = 8 mice/group). (C, D) Tumor volume was measured by digital caliper and plotted vs. time for each treatment cohort. (E, F) Kaplan–Meier survival curves of mice enrolled in the different treatment groups. Mice were treated for 21 days and followed for up to 40 days or until tumors reached a pre-defined volume (1,500 mm³). Error bars represent SEM.

Data information: *P*-values were calculated using Student's *t*-test (panels B–D) or log-rank Mantel–Cox test (E, F).
Source data are available online for this figure.

BRAFi-resistant Yumm1.7-BR syngeneic mouse model with a fully competent immune system (Behera *et al*, 2017). The combination of OTX-015 and PD901 inhibited tumor growth rate and increased animal survival (Fig 6A and B, Appendix Table S4A). Additionally, the OTX-015/PD901 combination did not cause any significant detrimental effects on immune cells of lymphoid and myeloid lineage (Appendix Fig S9A–D).

Next, we tested the efficacy of combining BET and MEK inhibitors in BRAFi-resistant patient-derived xenograft (PDX) models. The BRAFi-resistant tumor, WM3936-1, was derived from a BRAF^V600E mutant patient who developed a subcutaneous lesion harboring *de novo* NRAS (Q61K) and PI3KCA (H1047Y) mutations after 9 months of treatment with the BRAFi dabrafenib (Krepler *et al*, 2016). Consistent with our previous results, OTX-015 alone did not prevent

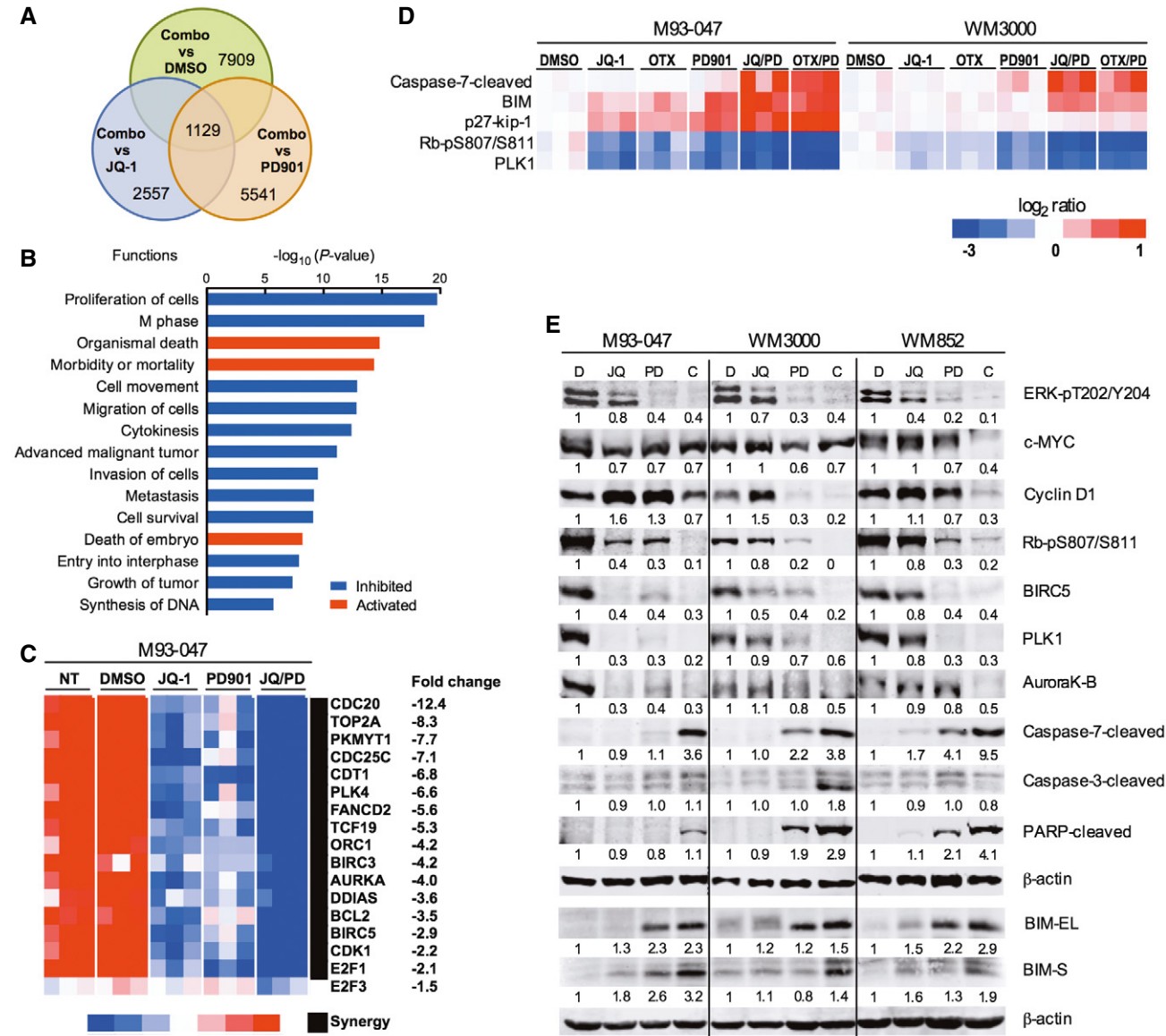

**Figure 4. BET and MEK inhibition synergistically impair cell cycle regulation and activate apoptotic signaling.**

A   Venn diagram of genes that were up- or downregulated after treatment of NRAS-mutant M93-047 cells with 0.5 μM JQ-1, 0.1 μM PD901, or combination (JQ-1/PD901) for 48 h ($P < 0.05$, FDR < 5%).

B   Ingenuity pathway analysis of functions significantly deregulated ($P < 0.05$) by combination treatment with JQ-1/PD901 for 48 h.

C   Heat map representation of the common cluster of differentially expressed genes ($n = 1,129$ genes) from 12 samples treated with DMSO, JQ-1, PD901, or combination for 48 h.

D   RPPA was performed in M93-047 and WM3000 cells treated with BETi (JQ-1 or OTX) or MEKi PD901 for 48 h as single agents or in combination. Analysis was performed in three biological replicates.

E   Immunoblotting in a panel of NRAS-mutant melanoma cells (M93-047, WM3000, and WM852) treated with vehicle, JQ-1, PD901, or combination for 48 h.

Source data are available online for this figure.

tumor growth, and PD901 transiently decreased the rate of tumor growth compared to vehicle and OTX-015 (Fig 6C and Appendix Table S4B). In contrast, the combination of OTX-015 and PD901 elicited significant tumor growth inhibition compared to monotherapy. Moreover, this combination significantly increased animal survival (Fig 6D) without apparent weight loss or signs of organ toxicity (Fig EV5). We performed additional studies in a syngeneic model derived from a conditional knock-in allele LSL-$NRAS^{Q61R}$ mouse model (Burd et al, 2014). Interestingly, this model responds poorly to anti-PD-1 treatment and is resistant to BETi or MEKi monotherapy (Fig 6E and Appendix Fig S9E). In contrast, co-targeting BET and MEK led to a striking reduction in tumor volume, improved animal survival, and decreased expression of TCF19 (Fig 6E and F, and Appendix Fig S9F).

Since TCF19 levels were associated with sensitivity to BET inhibitors (Fig 5B) and patient's survival (Fig 5E), we examined the expression of TCF19 in serial biopsies (pre-treatment, on-treatment, and progression) derived from patients treated with various targeted (BRAFi/MEKi) and/or immunotherapies (e.g., anti-CTLA, anti-PD1, anti-PDL1, IL-2; Appendix Table S5). TCF19 mRNA levels were downregulated in 93% post-treatment biopsies from responders and upregulated in 82% of progression biopsies from non-responders

(Fig 6G; Fisher's exact P = 0.001). Overall, these data raised the possibility that tumors resistant to targeted and/or immunotherapies expressing high levels of TCF19 could be treated with BETi/MEKi combinations. Indeed, the combination of OTX-015/PD901 induced substantial apoptosis of short-term cultures derived from immunotherapy-resistant patients (Appendix Table S6), along with inhibition of TCF19 and activation of apoptotic markers (Fig 6H and Appendix Fig S9G).

Altogether our data demonstrate that co-targeting BET and MEK elicits potent anti-tumor effects in highly drug-resistant NRAS-mutant melanoma models and support the premise that BETi/MEKi combinations may be a valuable salvage strategy for MAPK and immune checkpoint inhibitor-resistant tumors (Fig 7).

# Discussion

Epigenetic readers, such as BET family proteins, play a key role by binding to chromatin and regulating the transcription of oncogenes and tumor suppressors. We report that BET proteins are overexpressed in NRAS-mutant melanoma, and that these tumors are dependent on the BET family member, BRD4. Likewise, the BET family

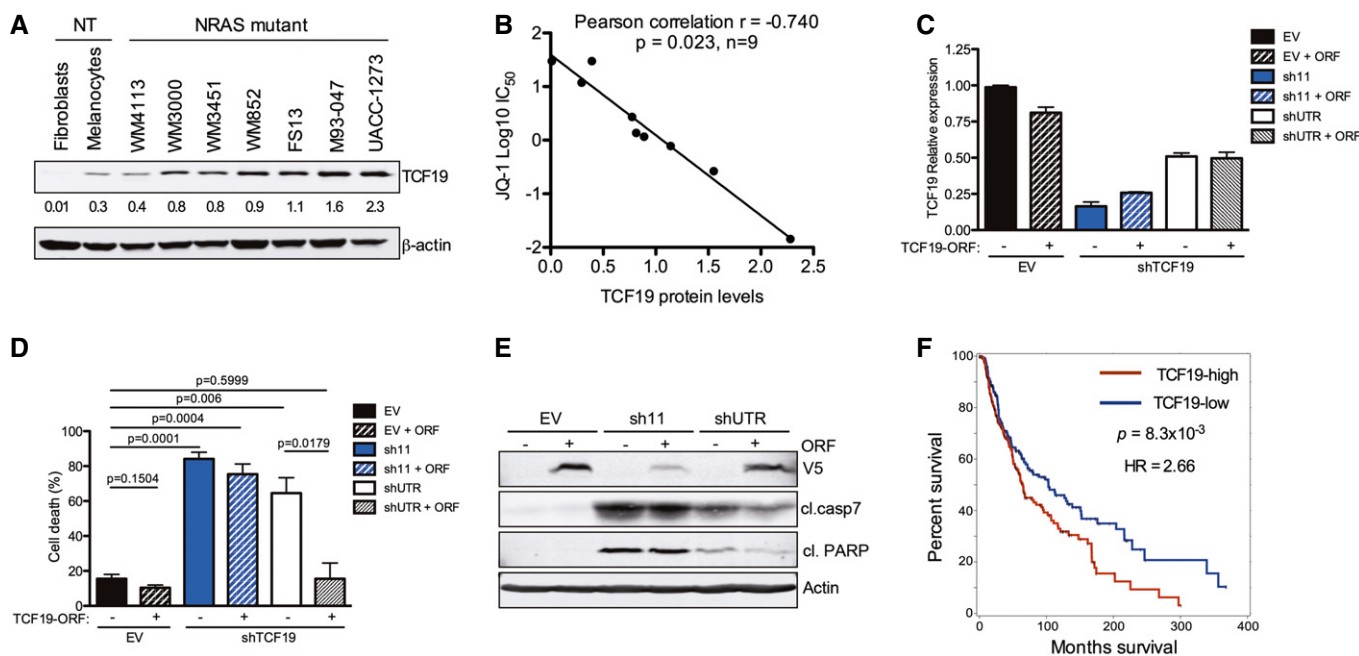

**Figure 5.  TCF19 is required for survival of NRAS-mutant melanoma cells.**

A    Expression of TCF19 was evaluated by Western blot in a panel of NRAS-mutant melanoma and non-transformed cells. Membranes were scanned and quantified using the LI-COR Odyssey system; β-actin was used as loading control. Quantification of TCF19 protein levels from immunoblots is shown below each band.

B    The linear correlation between TCF19 protein levels and JQ-1 IC$_{50}$ was assessed using Pearson's correlation coefficient. Data analysis was performed using Stata version 13.

C–E   TCF19 in NRAS-mutant WM852 melanoma cells was silenced using two different hairpins (sh11 targets the TCF19 coding region, and shUTR targets the 3′ UTR). EV: empty vector control. ORF: TCF19 open reading frame. (C) RNA was extracted, and mRNA levels were quantified by qRT–PCR. Data shown are the average of three replicates ± SEM. (D) Cell death was analyzed by flow cytometry using Annexin V and propidium Iodide 9 dpi. Percent (%) cell death (Annexin V$^+$/PI$^+$ cells) is shown. Data represent the mean of three independent experiments ± SEM; P-values were calculated by Student's t-test. (E) Cell lysates (4 dpi) were analyzed by immunoblotting with the indicated antibodies. Actin was used as loading control; antibodies against V5 were used to detect ectopic TCF19 (V5-tagged TCF19 ORF).

F    Skin cutaneous melanoma samples (n = 463) from the TCGA database (TCGA, SKMCC Provisional 2017) were stratified into two groups according to the median tissue TCF19 mRNA expression level (TCF19-low, n = 232 and TCF19-high, n = 231). Overall survival curves for TCF19 are shown; P-value and hazard ration (HR) were calculated by Cox regression comparing the two groups.

Source data are available online for this figure.

member BRD2 has been shown to have oncogenic potential and cooperate with RAS (Denis *et al*, 2000; Greenwald *et al*, 2004). In melanoma, BRD2 interacts with the histone variant H2A.Z.2, leading to upregulation of cell cycle genes and proliferation (Vardabasso *et al*, 2015). These findings are consistent with previous studies whereby BET inhibition restrained the growth of BRAF^V600E melanoma cells (Segura *et al*, 2013; Gallagher *et al*, 2014; Paoluzzi *et al*, 2016; Fallahi-Sichani *et al*, 2017). Together these reports support the notion that BET/BRD proteins play a critical role in melanoma and constitute promising therapeutic targets. We found that genetic silencing of

BRD4 or pharmacological inhibition of BET/BRD proteins has mainly cytostatic effects in NRAS-mutant melanoma cells *in vitro* and minimal effects *in vivo*. However, BETi/MEKi combinations elicit robust anti-tumor effects in NRAS-mutant melanoma.

Co-targeting BET and MEK lead to synergistic downregulation of multiple genes that stall the initiation of DNA replication and G2/M cell cycle checkpoint. Our results are consistent with studies in malignant peripheral nerve sheath tumors (MPNST), where a set of 25 genes were similarly downregulated by BETi/MEKi (De Raedt *et al*, 2014). Likewise, combining BRAF and BET inhibitors elicited

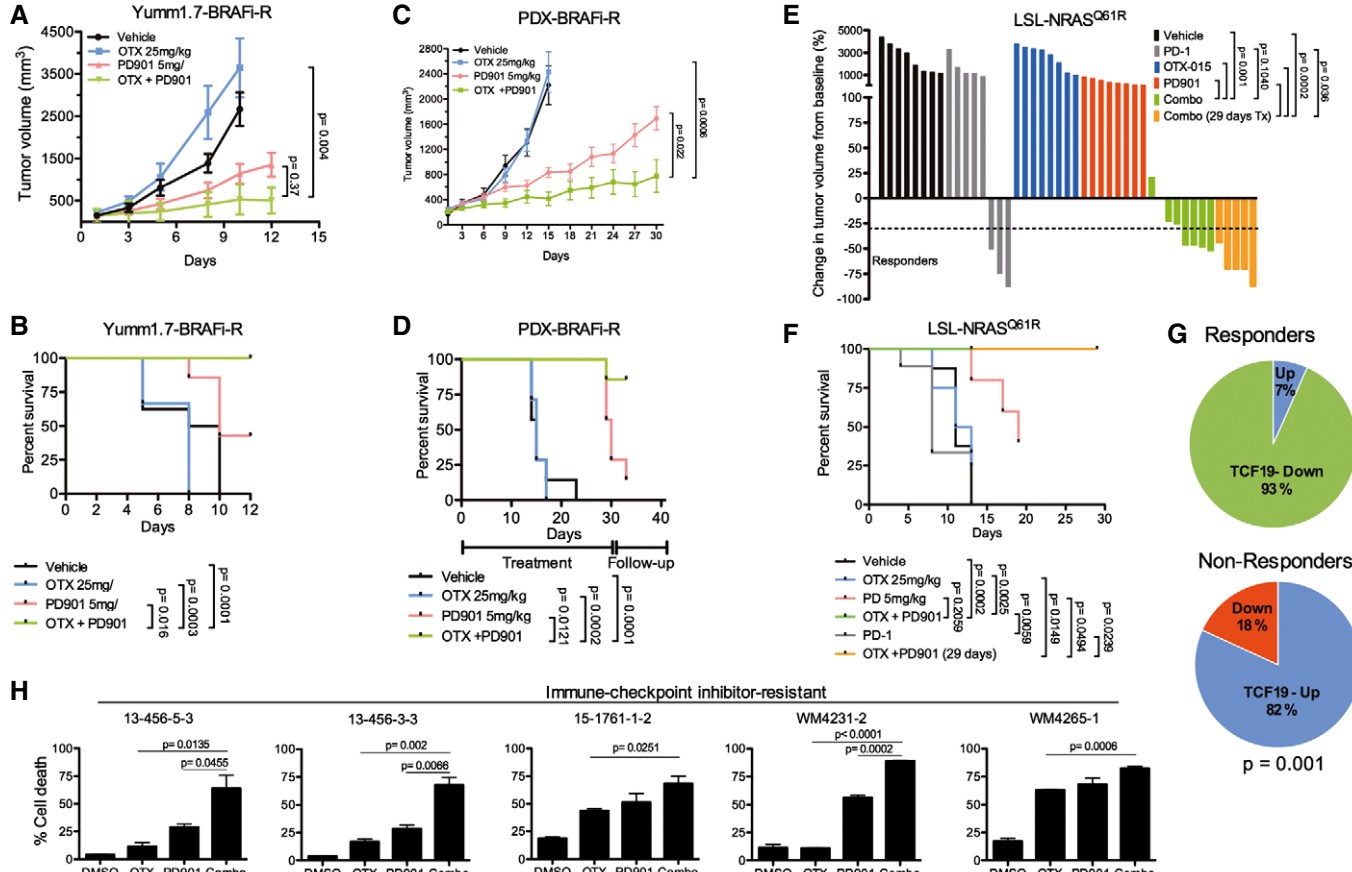

**Figure 6.  Combined BET/MEK inhibition offsets therapy resistance.**

A, B  BRAFi-resistant Yumm1.7-BR syngeneic tumors (*n* = 8 mice/group). Mice were treated for up to 12 days, treatment withdrawn and mice followed up until tumors reached a pre-set volume (1,200 mm³). (A) Tumor volume. (B) Kaplan–Meier survival curves.

C, D  WM3936-1 BRAFi-resistant PDX. Mice (*n* = 8 mice/group) were treated for up to 30 days or until tumors reached a pre-set volume (1,500 mm³). (C) Tumor volume. (D) Kaplan–Meier survival curves.

E, F  C57Bl/6 mice (*n* = 8 mice/group) bearing LSL-NrasQ61R/Q61R (TpN61R/61R) syngeneic tumors were treated as indicated. Mice were treated for up to 29 days or until tumors reached a pre-set volume (900 mm³). Mice were given three doses of anti-PD-1 (300 µg) every 5 days. (E) Waterfall plot depicting change in tumor volume from baseline for each mouse. Horizontal dashed line indicates 30% decrease in tumor volume. (F) Kaplan–Meier survival curves of mice enrolled in the different treatment groups. Data representative of two independent experiments.

G  Change in expression levels of TCF19 between pre-treatment and post-treatment biopsies was determined in a cohort of 23 patients that were treated with targeted therapy or immunotherapy. Non-responder patients include patients with progressive disease (*n* = 11 post-treatment biopsies from nine patients), and responders include patients with stable disease, partial response, or complete response (*n* = 15 post-treatment biopsies from 14 patients).

H  Short-term cultures derived from immunotherapy-resistant patients were treated as indicated for 7 days. Cells were stained with propidium iodide and Annexin V-FITC and analyzed by FACS; % dead cells (Annexin V^+/PI^+ cells) are shown.

Data information: Data in (A, C, H) represent the mean of three independent experiments ± SEM. In (A–F), mice bearing tumors were randomized into different treatment groups as indicated. *P*-values were calculated using Student's *t*-test (A, C, H), log-rank Mantel–Cox test (B, D, and F), Fisher's test (G), and Barnard's exact test (E).

Source data are available online for this figure.

synergistic anti-tumor effects by in BRAF-mutant melanoma (Paoluzzi *et al*, 2016). The combination of BET and BRAF inhibitors downregulated genes related to the cell cycle and DNA replication, and induced apoptosis through intrinsic and extrinsic pathways. Our findings indicate that treatment of NRAS-mutant melanoma with BET/MEK inhibitors blocks multiple stages of the major cell cycle transitions during cell division. Notably, we found that concurrent BET and MEK inhibition synergistically downregulates TCF19. Our results raise the possibility that the synergistic repression of the transcription factor TCF19 triggered by concomitant BET and MEK inhibition renders NRAS-mutant melanoma cells more vulnerable to apoptosis. Thus far, very little is known about the role and regulation of TCF19. It has been reported that TCF19 could play a role in the transcription of genes required for cell cycle progression and apoptosis of pancreatic beta cells (Krautkramer *et al*, 2013). Studies

in gastric cancer and insulinoma cells have shown that TCF19 regulates the expression of G1 (e.g., cyclins A and E) and G2/M (cyclin B) cyclins (Miao *et al*, 2013), and BIM (Krautkramer *et al*, 2013), but its role in melanoma is not known. Of note, the TCF19 promoter contains putative TF binding sites for AP1, GATA1,2,3, and p300 which can be regulated by MEK/ERK and BET. Depletion of TCF19 led to melanoma cell death, and TCF19 expression levels were associated with poor patient survival. Additionally, combined BET/MEK inhibition synergistically downregulated TCF19 triggering death of therapy-resistant tumor cells. Collectively, these findings suggest that BETi/MEKi synergistically block the initiation of DNA replication, prompt TCF19 downregulation coupled to inhibition of cell cycle checkpoints, leading to robust transcriptional perturbation and activation of pro-apoptotic signaling. Hence, our results indicate that targeting general epigenetic regulators and/or transcriptional

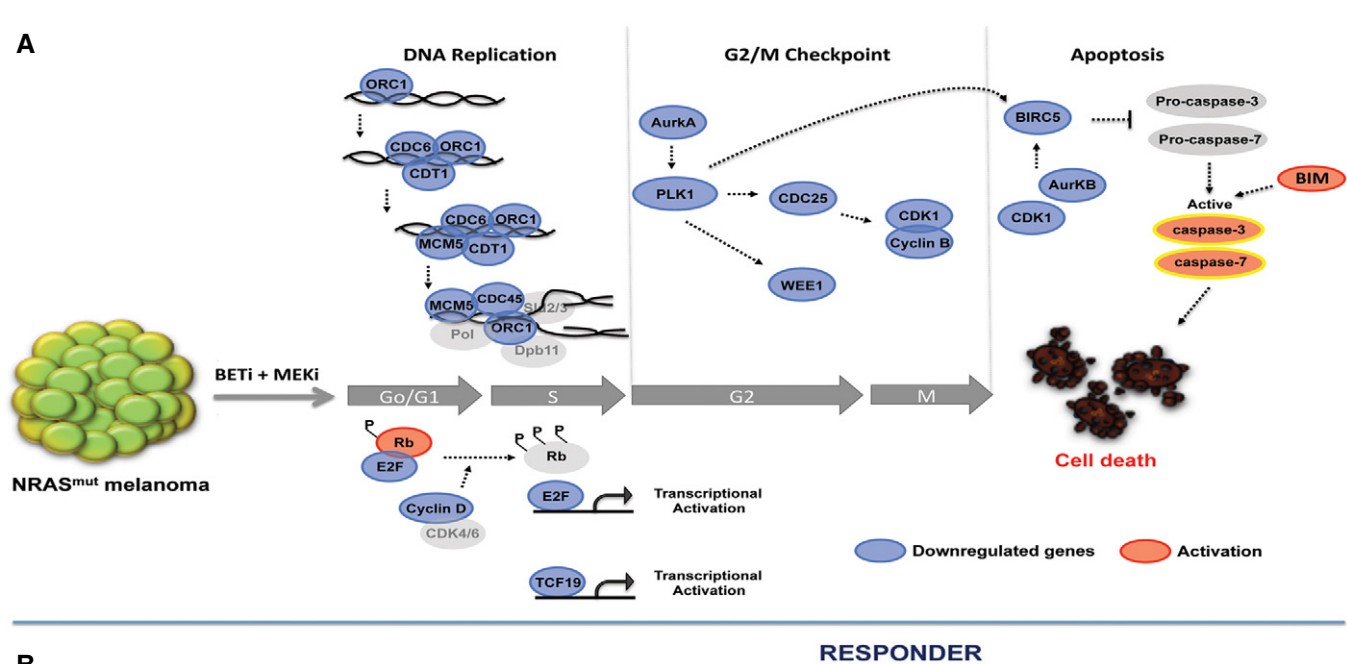

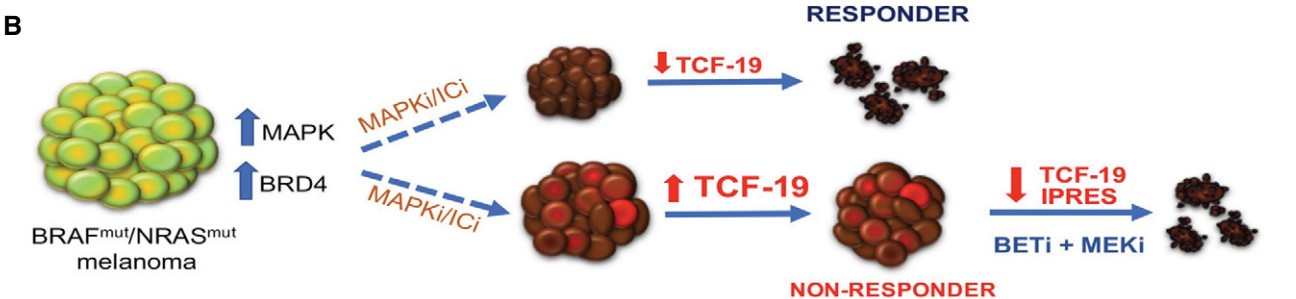

**Figure 7. Model depicting the molecular events leading to apoptosis of melanoma cells upon combined BET/MEK inhibition.**

A Targeting BET and MEK leads to downregulation of cyclin D, hypo-phosphorylation of Rb, and inhibition of E2F-dependent genes required for G1-S transition. The combination treatment downregulates the pre-initiation complex (ORC1, Cdc6, Cdt1, MCM5) impairing DNA replication in S phase. Inhibition of the G2/M checkpoint control via synergistic downregulation of Aurora kinase A, Plk1, Cdc25, CDk1, cyclin B, and Wee1 induces cell cycle arrest of NRAS-mutant melanomas. Repression of PLK1, CDK1, and Aurora kinase B blocks BIRC5, prompting activation of caspase-3 and caspase-7. Suppression of TCF19 and increased levels of BIM also contribute to activation of effector caspases and induction of cell death in NRAS-mutant melanoma cells (Blue: downregulation; Red: activation).

B BRAF and/or NRAS-mutant melanoma are generally treated with MAPK (MAPKi) or immune checkpoint inhibitors (ICi). In responder tumors, TCF19 is downregulated. In non-responder tumors, TCF19 levels are not affected. Treatment of MAPKi/ICi-resistant tumors with BETi/MEKi downregulates an anti-PD1 resistance innate immune signature and TCF19 leading to tumor cell death.

coactivators can produce remarkable anti-tumor effects with minimal toxicity to normal tissues.

While treatment of metastatic melanoma has been dramatically transformed by new targeted and immune therapies, these treatments are hampered by rapid onset of drug resistance. An added problem is the number and heterogeneity of resistance mechanisms, which can be mediated by genetic or epigenetic alterations (Shi et al, 2014). Recent studies have shown that transcriptional plasticity and acquisition of a mesenchymal phenotype can render tumor cells drug tolerant (Hugo et al, 2016). We noted that BETi/MEKi mitigated a transcriptional signature termed IPRES and identified in pre-treatment samples of patients intrinsically resistant to MAPK or immune checkpoint inhibitors (Hugo et al, 2016). We found that BETi/MEKi repressed the top gene sets in this signature, including genes regulating cell migration and extracellular-matrix remodeling, angiogenesis, and hypoxia, raising the possibility that targeting these processes may offset drug resistance. Indeed, using multiple models, we show that BETi/MEKi restrains both MAPK and immune checkpoint inhibitor-resistant tumors. These data suggest that combination treatment with BET and MEK inhibitors could be translated as an efficacious salvage approach for tumors resistant to MAPK and immune checkpoint inhibitors. Our data also indicate that the BETi/MEKi combination is well tolerated and does not cause significant detrimental effects on immune cells, consistent with preliminary clinical data indicating that side effects secondary to BET inhibitors are manageable; no dose-limiting toxicities have been reported in doses up to 80 mg once a day (Berthon et al, 2016). Thus far, reported toxicity appears to be reversible and includes diarrhea and fatigue at doses higher than 160 mg in patients with acute myeloid leukemia, and thrombocytopenia, at doses higher than 80 mg in patients with other hematologic malignancies (Amorim et al, 2016).

While uncovering specific mechanisms of drug resistance is certainly necessary, identifying strategies to overcome diverse and heterogeneous mechanisms of therapy resistance is crucial. Our studies have uncovered BRD4 as a key vulnerability in NRAS-mutant melanoma and demonstrate that co-targeting BET and MEK could be a highly efficacious strategy to treat melanoma refractory to current therapies. Our findings also suggest that high BRD4 and TCF19 expression are associated with increased response to BET inhibitors, raising the possibility that BRD4/TCF19 levels could be used as markers to select patients most likely to benefit from this therapy. However, this needs to be further investigated in prospective studies. Considering that BET inhibitors are in advanced clinical trials, and the MEK inhibitor trametinib is FDA-approved for BRAF-mutant melanoma patients, clinical studies combining BETi and MEKi could be rapidly implemented for therapy-resistant melanoma, assessing a promising treatment strategy that could improve the outcomes of patients who have failed all currently available therapies.

# Materials and Methods

### Cell lines, reagents, and viability assays

The Wistar melanoma (WM) cell lines were established in the Herlyn laboratory at The Wistar Institute. M93-047 cells were described previously (Bittner et al, 2000). All melanoma cells were propagated in RPMI-1640 medium (Thermo Scientific, Logan, UT), supplemented with heat-inactivated 5% fetal bovine serum (FBS; Thermo Scientific), and maintained at 37°C in 5% $CO_2$. All cell lines were authenticated by DNA fingerprinting using Coriell microsatellite kit (Camden, NJ) and screened for Mycoplasma using MycoAlert Mycoplasma Detection Kit (Lonza Inc., Allendale, NJ). For short-term cultures, tumor cells were isolated from fresh tumor biopsies performed on patients. Samples were received de-identified, and tumor cells were maintained in RPMI supplemented with 10% heat-inactivated FBS. Fibroblast outgrowth in short-term cultures was eradicated using differential trypsinization and geneticin. The purity of tumor cell population was confirmed by S100 staining.

Small molecule inhibitors were purchased from Selleckchem (Houston, TX; PD0325901 (S1036), PD033991 (S1579), BKM-120 (S2247), or MedChemExpress (Monmouth Junction, NJ; OTX-015 (HY-15743). JQ-1 was a generous gift of Dr. James Bradner (Harvard University; current affiliation Novartis Institutes for BioMedical Research (NIBR), Cambridge, MA). Cell viability was measured using 3-(4,5-dimethylthiazol-2-yl)-2,5-diphenyltetrazolium bromide or resazurin (Biosynth International, Itasca, IL). Concentrations inhibiting 50% of cell growth ($IC_{50}$) were calculated after three or 6 days of drug treatment using GraphPad Prism V5.0a.

### Analysis of publicly available datasets

TCGA skin cutaneous melanoma dataset (SKCM, Provisional 2017) was accessed via www.cbioportal.org (Cerami et al, 2012; Gao et al, 2013). mRNA expression of BRD4, BRD3, and BRD2 for NRAS ($n = 98$), BRAF ($n = 188$), NF1 ($n = 47$) mutant, and WT/WT/WT ($n = 170$) melanoma samples were downloaded and stratified into two groups (BRD-low or BRD-high) according to the median tissue mRNA expression levels. Subsequently, mRNA expression for BRD4, 3, and 2 was plotted using GraphPad Prism V5.0a. Correlation between gene of interest (BRD4, 3, or 2) and overall survival or disease-free survival was determined using the Kaplan–Meier method. Differences in survival curves were determined using the log-rank test.

Data from 12 patients treated with BRAF or BRAF/MEK inhibitors were retrieved from the European Genome-phenome archive (accession number EGAS00001000992; https://www.ebi.ac.uk/ega/studies/EGAS00001000992; Kwong et al, 2015).

### Interaction index

To determine synergy, interaction index was calculated for every combination using a dose–response surface model based on Bliss independence principle (Harbron, 2010; Liu et al, 2018). Interaction index ($\tau$) and 95% confidence interval (CI) were calculated to evaluate the combination effect of two drugs. For combinations with two given drugs, when the $\tau < 1$ and the upper limit of its 95% CI is also less than 1, the combination effect of the two drugs was considered as significant synergism.

### Apoptosis assay

Treated cells were trypsinized, washed with Annexin binding buffer (0.14 M NaCl, 2.5 mM $CaCl_2$, 0.1 M Hepes, pH 7.4), and labeled with 2.5 μl Annexin V-APC (8 μg/ml; BD Biosciences) for 30 min at room temperature. Subsequently, samples were labeled with 1 μl PI

(1 mg/ml, Sigma-Aldrich) for 10 min at room temperature. Apoptotic (Annexin $V^+$/$PI^+$) cells were quantified by FACS (LSRFortessa, BD Biosciences) and analyzed using FlowJo software (v.10.0.8).

### 3D melanoma spheroids

3D melanoma spheroids were generated as previously described (Villanueva et al, 2010; Shannan et al, 2016). Briefly, 5,000 melanoma cells were seeded onto 96-well culture plates coated with 50 μl of 1.5% agar and incubated at 37°C, 5% $CO_2$ for 3 days. The spheroids were embedded into a collagen mixture (10× EMEM, 200 mM L-glutamine, FBS, 1.5 mg/ml collagen, and 7.5% $NaHCO_3$) and treated with 0.5 μM BETi, 0.1 μM MEKi, or combination of both drugs. Five days later, spheroids were washed with PBS and stained with Calcein (AM) (eBioscience) and ethidium bromide (Sigma-Aldrich). Spheroids were visualized and imaged using an inverted fluorescence microscope (Nikon TE2000, 4× objective).

### Animal models and therapy studies

All studies and procedures involving mice were approved by the Institutional Animal Care and Use Committee (IACUC) at the Wistar Institute (protocol number 112584), performed in an Association for the Assessment and Accreditation of Laboratory Animal Care (AAALAC) accredited facility, and conform to all regulatory standards. NOD.Cg-Prkdcscid Il2rgtm1Wjl/SzJ (NSG) and C57Bl/6 mice were obtained from The Wistar Institute or Charles River, respectively. Male and female mice from 4 to 6 weeks old were used. Animals were housed in groups of five mice/cage in a AAALAC-certified facility. Mice were maintained and handled in accordance with the Wistar Institute IACUC.

#### Tumor xenografts
M93-047 ($1 \times 10^6$) cells were suspended in matrigel (500 μg/ml) and injected subcutaneously into 4- to 6-week-old female or male NSG (NOD.Cg-Prkdc$^{scid}$ Il2rg$^{tm1Wjl}$/SzJ) mice.

#### Patient-derived xenografts
PDX models have been characterized and were established as previously described (Krepler et al, 2016). Briefly, frozen tumors were washed once with RPMI-1640 and minced. Subsequently, animals were anesthetized by inhalation of 3% isoflurane, and their skin was cleaned with alcohol. A small skin incision was made in the lateral flank and minced tumor fragments mixed with 100 μl of matrigel were inserted (BD Biosciences, San Jose, CA, USA). The incision site was closed with surgical clips, which were removed after 7 days. Mice were monitored every day. The established xenografts were subsequently expanded.

#### Syngeneic mouse models
YUMM1.7-BRAFi-R cells were derived from the BRAFV600E-PTEN$^{L/L}$ mouse model (Dankort et al, 2009) and established as described (Meeth et al, 2016; Behera et al, 2017). Yumm1.7-BRAFi-R cells ($2.5 \times 10^5$) were suspended in Matrigel and injected subcutaneously into C57Bl/6 mice (Charles River). Syngeneic NRAS-mutant tumors were derived from the Tyr-CRE-ERT2; p16$^{L/L}$; LSL-NrasQ61R/Q61R (TpN61R/61R) mouse model (Burd et al, 2014). TpN61R/61R tumors were excised, washed with RPMI-1640, minced,

and implanted into C57Bl/6 mice (Charles River) as described (Burd et al, 2014; Krepler et al, 2016).

#### Drug treatments
Tumor-bearing mice were randomized into different treatment groups: vehicle, JQ-1 25 mg/kg; ip.qd., OTX-015 25 mg/kg ip.qd., PD0325901 5 mg/kg ip.qd., or combination (JQ1/OTX-015 + PD901). Mice were treated with three doses of anti-PD1 (3 mg/kg) given i.p. every 5 days. Mice were weighed twice a week and tumor growth was measured every other day with digital calipers. Tumor volume was calculated using the formula: tumor volume ($mm^3$) = (length (mm)) × (width (mm)$^2$) × 0.5.

### Histological analysis

Organs (lung, liver, spleen, and kidney) from treated mice were collected, fixed with 4% paraformaldehyde solution, and embedded in paraffin. Organ sections of 5 mm thickness were stained by hematoxylin and eosin (H&E) and examined by optical microscopy (Eclipse E600, Nikon Instruments Inc., Melville, NY).

### Analysis of immune cells in immunocompetent syngeneic mouse model

Splenocytes and tumor cells were purified using the Miltenyi tumor dissociation kit (cat no. 130-096-730, Miltenyi Biotec, San Diego, CA). Tumors were excised and cut into small pieces using surgical blades. Small pieces were homogenized with collagenase and master mix enzymes R and A, and placed on a tumor homogenizer (protocol 02) for one cycle. The homogenized tumor was incubated for 30 min at 37°C in a tube rotator (Miltenyi MacsMix). After incubation, cells were washed and resuspended in 10 ml MACS buffer (1× PBS supplemented with 0.5% fetal calf serum and 2.5 mM EDTA). Spleens were homogenized using a syringe plunger and passing cells through a 70-μm cell strainer. Subsequently, cells were washed with MACS buffer, incubated for 1 min with ACK buffer, and resuspended in MACS buffer.

Cells were stained with antibodies for intracellular and cell surface markers and analyzed by flow cytometry. Dead cells were excluded by co-staining with DAPI (1:1,000 dilution; Sigma, St Louis, MO; D9542) and Zombie Yellow (Biolegend, San Diego, CA; 423103; 1:1,000 dilution). Antibodies for cell surface markers used were as follows: APC anti-mouse CD45 (103112), APC/CY7 anti-mouse CD11C (117323), Alexa flour 700 anti-mouse Ly-6G (127621), Alexa flour 700 anti-mouse CD3 (100216), PE anti-mouse CD8a (100707), Brilliant violet 510 anti-mouse CD4 (100553), PE/Dazzle 594 anti-mouse CD152 (106317), PerCp/Cy5.5 anti-mouse/human CD45R/B220 (103235; Biolegend, San Diego, CA), anti-mouse CD11b PE-cyanine 7 (25-112-81) and anti-mouse F4/80 antigen PE (12-4801-80; eBioscience, Gran Island, NY), and FITC rat anti-mouse Ly-6C (BD Biosciences, San Jose, CA; 561085). FOXP3 was stained with pacific blue anti-mouse antibody (Biolegend, San Diego, CA; 126409). All antibodies were used at 1:100 dilution.

### Immunohistochemical analysis of patient tumor samples

Formalin-fixed, paraffin-embedded (FFPE) sections of patient samples from NRAS-mutant ($n = 18$), BRAF-mutant ($n = 19$), and

WT/WT ($n = 17$) metastatic melanoma patients ($n = 54$) were selected from the surgical pathology files at the University of Pennsylvania Medical Center. The protocol was approved by University of Pennsylvania Institutional Review Board (IRB). Immunohistochemistry of FFPE samples was performed using antibodies against human BRD4 (Bethyl laboratories, Inc; A301-985A50). Staining was done on a Leica Bond-IIITM instrument using the Bond Polymer Refine Red Detection System (Leica Microsystems DS9390). Heat-induced epitope retrieval was done for 20 min with ER1 solution (Leica Microsystems AR9961). Incubation with the anti-BRD4 antibody (1:2,000) for 15 min, followed by 20-min post-primary AP and 30-min. incubation with polymer AP. Immunoreactivity of nuclear BRD4 staining was blindly analyzed by a board-certified dermatopathologist (X.X.) using the semiquantitative four-tiered scale (*H*-Score method) with 0–3 corresponding to negative staining (0), weak (1), intermediate (2), and strong staining (3). The percentage of positive cells and staining intensity was analyzed by visual assessment, and the *H*-score calculated using the formula $1 \times$ (% of weak staining nuclei) + $2 \times$ (% of intermediate staining nuclei) + $3 \times$ (% of strongly staining nuclei), giving a range of 0–300.

## Patient tumor samples

Tumor biopsies were collected from melanoma patients under IRB-approved protocols at the Massachusetts General Hospital (DF-CI 11-181), the Hospital of the University of Pennsylvania (703001), and the Wistar Institute (2802240). Patients were treated with targeted (BRAF/MEK inhibitors) or various immunotherapies. All patients provided written consent for tissue acquisition and analysis, and studies conformed the principles set out in the WMA Declaration of Helsinki and the Department of Health and Human Services Belmont Report.

## Statistics

For all *in vitro* experiments, unless otherwise indicated, data are presented as the mean ± SEM of three independent experiments. Significant differences between experimental conditions were determined using a 2-tailed, unpaired Student's *t*-test. For tumor volume analysis, Student's *t*-test with unequal variances was used to determine the differences in average tumor growth rates between treatment groups. Barnard's exact test was used to determine the differences in percent tumor response rate between treatment groups. For survival analysis, Kaplan–Meier survival curves were generated, and their differences were examined using the log-rank test. A two-tailed *P*-value of < 0.05 was considered statistically significant.

## Data availability

RNA-sequencing data from this publication have been deposited to Gene Expression Omnibus and assigned the identifier accession number GSE95153 (https://www.ncbi.nlm.nih.gov/geo/query/acc.cgi?token=wxydcyeezbknbcx&acc=GSE95153).
RPPA data from this publication have been deposited in FigShare (https://figshare.com/s/ede29446d6ad2d124727).
The patients' dataset used in this publication was retrieved from the European Genome-phenome archive (accession number EGAS00001000992; https://www.ebi.ac.uk/ega/studies/EGAS00001000992).

**Expanded View** for this article is available online.

**The paper explained**

**Problem**
While treatment of melanoma has been transformed by new targeted and immunotherapies, thus far there are no approved targeted therapies for nearly 30% of melanomas harboring NRAS mutations. An added problem is that most treatments are hindered by the rapid onset of drug resistance. Moreover, there are no effective salvage therapies available for patients who fail targeted and immune therapies.

**Results**
We found that the epigenetic regulator BRD4 is expressed at high levels in NRAS$^{Mut}$ melanoma and that BRD4 is required for tumor cell viability. Furthermore, high levels of BRD4 are associated with poor outcome in NRAS$^{Mut}$ melanoma patients. We demonstrate that co-targeting BET and MEK synergistically restrains tumor growth and prolongs the survival of NRAS$^{Mut}$ tumor-bearing mice with no overt toxicity. Transcriptomic analysis revealed that co-treatment with BET and MEK inhibitors mitigated a transcriptional signature associated with innate resistance to immune checkpoint and targeted inhibitors. Accordingly, BETi/MEKi combinations inhibited the growth of anti-PD1- and BRAFi/MEKi-resistant tumors. Moreover, this combination was highly efficacious and well tolerated in both immunocompetent syngeneic mouse models and patient-derived xenografts. We further discovered that co-targeting BET and MEK downregulates TCF19 and that this transcription factor is required for melanoma cell survival. Analysis of tumor samples from patients treated with targeted or checkpoint inhibitors suggests that downregulation of TCF19 is associated with therapy response.

**Impact**
Our studies have uncovered BRD4 as a key vulnerability in NRAS-mutant melanoma and establish that co-targeting BET and MEK may be an effective strategy that could be rapidly translated for the treatment of melanoma refractory to current therapies.

## Acknowledgements

We would like to thank James Hayden and Frederick Keeney (Imaging Core Facility), Jeffrey S. Faust (Flow Cytometry), Celia Chang (Genomics Facility), and Denise DiFrancesco (Animal Core Facility) at the Wistar Institute for technical support. We thank Katrin Sproesser for providing PDX tumor samples and cell lines, Curtis Kugel for assistance with analysis of immune cells, Minu Samanta and Elene Tsopurashvili for technical assistance with immunoblots, and Regina Stoltz for assistance organizing de-identified patient data. JQ-1 was a kind gift from Dr. James Bradner (Harvard University). We are grateful to Maureen Murphy and Rugang Zhang (The Wistar Institute) for critical reading of the manuscript and Rachel Locke for editorial assistance. The RPPA analysis was performed by the MDACC RPPA core facility with support for shared resources provided by Cancer Center Support Grant (CCSG) CA016672 to MDACC. Support for RPPA was provided by the Dr. Miriam and Sheldon G. Adelson Medical Research Foundation to M. Herlyn. Support for shared resources utilized in this study was provided by Cancer Center Support Grant (CCSG) P30CA010815 to the Wistar Institute. Work in our laboratory is supported by NIH grants R01CA215733, K01CA175269, P01CA114046, P50CA174523, the American Cancer Society, the V Foundation for Cancer Research, the Melanoma Research Alliance, Melanoma Research Foundation,

and the Martha W. Rogers Trust. IEV was supported by NCI NRSA T32 CA009171 Cancer Biology Training Grant to the Wistar Institute.

## Author contributions

IME-V and JV conceived and designed the study, and wrote the manuscript. IME-V, PIR-U, and ANG performed experiments, collected, and analyzed data. XY and QL performed bio-statistical analysis. CC, GZ, and ZW performed IPRES ssGSEA analysis. AVK and ZW performed bioinformatics analysis. CK and MH provided PDX samples; AEA, ATW, MH, and RS provided cells, reagents or scientific input; and CEB provided syngeneic NRAS-mutant tumors. XX performed IHC, analyzed, and scored patient samples. RA, GK, WX, JJDM, LMS, RJS, GB, KTF, MB, and DTF provided patient samples. YL and GBM performed RPPA. All authors contributed, reviewed, and approved the manuscript.

## Conflict of interest

Gordon B. Mills serves as a consultant for AstraZeneca, Blend Therapeutics, Critical Outcome Technologies Inc., HanAl Bio Korea, Illumina, Nuevolution, Pfizer, Provista Diagnostics, Roche, SignalChem Lifesciences, Symphogen, Tau Therapeutics; owns stock in Catena Pharmaceuticals, PTV Healthcare Capital, Spindle Top Capital; and has received research funding from Adelson Medical Research Foundation, AstraZeneca, Critical Outcome Technologies Inc., GSK, and Illumina. All the other authors declare no conflict of interest.

## For more information

(i)  The Wistar Institute: https://www.wistar.org
(ii)  MRF: https://www.melanoma.org
(iii)  MRA: https://www.curemelanoma.org
(iv)  https://www.inspire.com/groups/melanoma-exchange/
(v)  ACS: https://www.cancer.org/cancer/melanoma-skin-cancer.html
(vi)  The V Foundation for Cancer Research: https://www.jimmyv.org
(vii)  Clinicaltrials.org

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
