## [Review Process File · EMBO Molecular Medicine]

Co-targeting BET and MEK as salvage therapy for MAPK and checkpoint inhibitor resistant melanoma

Ileabett M. Echevarría-Vargas, Patricia I. Reyes-Uribe, Adam Guterres, Xiangfan Yin, Andrew V. Kossenkov, Qin Liu, Gao Zhang, Clemens Krepler, Chaoran Cheng, Zhi Wei, Rajasekharan Somasundaram, Giorgos Karakousis, Wei Xu, Jennifer J.D. Morrissette, Yiling Lu, Gordon B. Mills, Ryan J. Sullivan, Miao Benchun, Dennie T. Frederick, Genevieve Boland, Keith T. Flaherty, Ashani T. Weeraratna, Meenhard Herlyn, Ravi Amaravadi, Lynn M. Schuchter, Christin E. Burd, Andrew E. Aplin, Xiaowei Xu, and Jessie Villanueva

Review timeline:

Submission date:	01 September 2017
Editorial Decision:	09 October 2017
Revision received:	2 February 2018
Editorial Decision:	22 February 2018
Revision received:	07 March 2018
Accepted:	12 March 2018

Editor: Céline Carret

Transaction Report:

1st Editorial Decision

09 October 2017

Thank you for the submission of your manuscript to EMBO Molecular Medicine. We have now heard back from the three referees whom we asked to evaluate your manuscript.

As you will see from the comments below, the three referees are enthusiastic about the study and do have suggestions and recommendations to further improve conclusiveness and clarity. I will not get into experimental details, but we feel that the referees' reports are very clear and nicely detailed and we would strongly encourage you to address all issues raised as recommended.

We would welcome the submission of a revised version within three months for further consideration and would like to encourage you to address all the criticisms raised as suggested to improve conclusiveness and clarity. Please note that EMBO Molecular Medicine strongly supports a single round of revision and that, as acceptance or rejection of the manuscript will depend on another round of review, your responses should be as complete as possible.

I look forward to receiving your revised manuscript.

***** Reviewer's comments *****

Referee #1 (Remarks for Author):

This study from Villaneuva and colleagues demonstrates that co-targeting the MAPK and BET pathways *in vivo* acts to impede both MAPKi and immunotherapy resistance in metastatic melanoma models. Acquired drug resistance in melanoma patients remains a major clinical challenge. Therefore, these findings are potentially clinically useful as a 'salvage therapy'. The authors also demonstrate MEKi and BETi work synergistically to inhibit growth of NRAS-mutant melanoma. As such, these important findings should be reported. The following points should be addressed prior to publication:

Major comments:

1. The authors focus on NRAS-mutant melanoma as a target for combined BET + MEK inhibition. However, they should also address the fact that a similar combination (BET + BRAFi) has also been shown to work *in vivo* for BRAF-mutant melanoma and cite the following: "BET and BRAF inhibitors act synergistically against BRAF-mutant melanoma", by Eva Hernando and colleagues (Cancer Medicine, 2016). On the same note, the authors should show the BET proteins as in Fig 1A in the other genetic subtypes (BRAF, NF1, WT/WT) in the Supp figures.
2. The authors claim that BRD4 is higher in NRAS-mutant melanoma tissues (n=9) vs. BRAF-mutant (n=8) and WT/WT (n=9), but do not use sufficient numbers to demonstrate this. The authors should perform IHC on additional FFPE tissues, which are likely available at their institute.
3. It is unclear why the authors chose to use PD901 as opposed to the clinically relevant MEKi, trametinib. Some of the assays should be repeated with trametinib, ie. Spheroids or *in vitro* proliferation assays. Moreover, the authors should perform Annexin V assays to determine the level of apoptosis using the MEKi + BETi combination.
4. The JQ1 sensitivity inversely correlates with BRD4 levels as shown in Supp Fig 1E. This very nice data should be moved to the main figures. Does this sensitivity also correlate with BRD2 levels? BRD2 has also been shown to play an important role in melanoma by Vardabasso et al., Mol Cell 2015. Moreover, JQ1 has shown cytostatic effects in an NRAS-mutant melanoma cell line in this paper as well and should be cited.
5. A model describing the findings should be incorporated into the main figures, particularly to understand why the combination works in both MAPKi-resistant and Immunotherapy-resistant cells. This is somewhat lost in the figures as the most of the IPRES data is in the Supp.

Minor comments:

1. The introduction does not provide sufficient details and citations of the role of epigenetics in melanoma.
2. The Supp figures seem to be listed out of order in the manuscript and can be confusing to follow.
3. For TCF19 expression, the authors should also look at the genetic subgroups (Fig 3D is all TCGA patients). Also, why does the KD of TCF19 induce apoptosis vs. cell cycle arrest? Is it Caspase-dependent?
4. LSL-NRAS model should be explained as a lox-stop-lox model. Were the injections of cells derived from the animals or cell lines? It is not clear.
5. Pharmacological inhibition of BRD4 alone does not exist; it is inhibition of all BETs, see discussion.
6. Some of the Supp figures are missing statistics.

Referee #2 (Remarks for Author):

The present study by Echevarria-Vargas and colleagues provides cogent evidence that the Bromodomain and Extraterminal Domain (BET) family protein member 4 (BRD4) is a therapeutic target in melanoma. BRD4 levels are elevated in NRAS mutant positive melanoma cells and high expression is associated with poor survival/disease progression. Knockdown/antagonism of BRD4 suppresses the proliferation and less so survival of melanoma cells. However, antagonism of MEK and BRD4 synergizes to kill cells both in 2d and 3d culture and in vivo. The potency of this combination is selective for melanoma cells, as it does not kill fibroblasts or immune cells and is well tolerated by mice. RNAseq and proteomics reveals the combination treatment affects cell cycle progression and survival networks and is mediated through transcription factor tcf19. The most exciting aspect of the study is the ability of MEKi/BDRi to suppress tumor growth of even MEKi and the proliferation of immune therapy resistant cells.

The study has been well performed and only the following suggestions are given to improve the study

Fig 1/S1 Define high and low expression of BET proteins (what percentile?) in the results text/figure legends

Fig 3. The extent of cell death induced by knockdown of tcf19 is not consistent with the degree of knockdown, raising concerns that the effect of shRNA#11 are not specific. To be convincing, the authors should use an additional shRNA sequence or attempt to suppress the phenotype with tcf19 overexpression.

Referee #3 (Remarks for Author):

The authors show that BRD4 expression is associated with poor patient survival in melanoma carrying mutated NRAS and suggested the BRD4 is a molecular target for NRAS mutant melanoma; knockdown of BRD4 and a small molecular BET inhibitor JQ1 affected the melanoma cells modestly in vitro and mildly in vivo. The authors then show that the combinations of BET and MEK inhibitors could inhibit the growth of NRAS-mutant melanoma and overcome the resistance to immune-therapy through downregulation of transcriptional factor TCF19. Overall, most of the data shown are convincing. Moreover, data demonstrating that co-targeting BET and MEK signalling blocked the cell proliferation of NRAS-mutant and immune-therapy resistant melanoma is a novel finding. However, the mechanism may need to be addressed.

Comments:

1) Novelty and mechanism. Although the demonstration of the combinations of BET and MEK inhibitors could inhibit the growth of NRAS-mutant melanoma and overcome the resistance to immune-therapy through downregulation of transcriptional factor TCF19 is novel, a role for BET or MEK inhibitor for treating melanoma is far from new. Is there any correlation of BRD4 and TCF19 expression and NRAS mutation in TCGA data? Does BRD4 regulate the TCF19? Does the combination affect the BRAF mutant melanoma? Or the authors should use the BRAF mutant melanoma as a control.

2) Page 5, the authors state that 'depletion of BRD4 decreased the viability of NRAS mutant melanoma cell, but induced only modest apoptosis', to have an explanation. Also, the degree of reduction of BRD4 does not correlate with apoptosis.

3) Figure 1 D lacks the label description of the box of light gray and dark gray.

4) In page 5, the authors state that 'single dose of either compound as a single agent transiently restrained cell proliferation, but this effect was not sustained at 14 days', to explain

5) In page 8-9, the authors state that 'depletion of TCF19 let to activation of caspase-7, PARP cleavage, and apoptosis of NRAS mutant melanoma cells (Fig. 3C)', the degree of reduction of TCF19 seems not correlate with caspase-7, PARP cleavage, and apoptosis of melanoma cell in Fig.3 C and D.

- 6) The sample size (cohort) in Figure 2 C to E is small (n=3). Cohort
- 7) In Figure G, the level of TCF19 expression should show (such as by PCR or IHC).

1st Revision - authors' response

02 February 2018

In this revised manuscript, we have addressed all of the concerns of the Reviewers. The result is a greatly strengthened manuscript with 8 new panels of data in the main figures and 10 new panels of data in the Supplemental figures. All three reviewers noted the potential therapeutic importance of targeting NRAS-mutant melanoma, which has limited available therapies, with this new combination of BET/MEK inhibitors. They also noted the clinical importance of targeting therapy-resistant melanoma. Below is a point-by-point response to the reviewers' comments.

POINT-BY-POINT RESPONSE:

Referee 1:

1. Reviewer 1 wanted us to include discussion of the work of Dr. Eva Hernando and colleagues on the BRAF/BET combination for BRAF mutant tumors. Along these lines, **Reviewer 2** wanted us to include data on BRAF mutant tumors, and Reviewer 1 requested an analysis of BRD proteins in other genotypes of melanoma.

Response: These are excellent suggestions and we have done this. We have cited the excellent paper by Dr. Hernando's group and we now include discussion of their data, and how it fits with ours, on p. 13.

Following this Reviewer's request, we also now include a new Supplementary figure (Appendix Fig. S1), showing overall survival and progression free survival in BRAF mutant, NF1 mutant, and WT/WT/WT melanoma patients, stratified based on expression of BET proteins, as suggested by the reviewer. In brief, analysis of the TCGA data showed that BET proteins are not significantly associated with patient outcome in BRAF mut, NF1 or WT/WT/WT genetic subtypes (Appendix Fig S1), so this association holds true only for NRAS-mutant melanoma. We thank these Reviewers for these suggestions.

2. *"The authors claim that BRD4 is higher in NRAS-mutant melanoma tissues (n=9) vs. BRAF-mutant (n=8) and WT/WT (n=9)...The authors should perform IHC on additional FFPE tissues ..."*

Response: We have done this. We have now analyzed 28 additional melanoma samples (9 NRASmut, 10 BRAFmut, and 9 WT/WT melanoma samples). The analysis and IHC scores in Figure 1B, now depicts values based on 18 NRASmut, 19 BRAFmut, and 17 WT/WT tumor samples derived from 54 patients. Importantly these data, including the additional samples, are completely consistent with our initial results, (new Fig 1C). We thank this Reviewer for this suggestion.

3. *"The JQ1 sensitivity inversely correlates with BRD4 levels as shown in Supp Fig 1E. This very nice data should be moved to the main figures. Does this sensitivity also correlate with BRD2 levels?" BRD2 has also been shown to play an important role in melanoma by Vardabasso et al., Mol Cell 2015...this paper as well and should be cited".*

Response: We have done this, and have moved the data in Supplementary Figure 1E to a new Figure 2. We do not find that BRD2 nor BRD3 levels correlate with sensitivity to JQ1 in NRAS mutant melanoma cells. We have added these data to the manuscript (New Fig EV2G and 2H). We thank the Reviewer for this suggestion and have cited the paper by Vardabasso *et al.* Pg. 13 of the revised manuscript.

4. *"It is unclear why the authors chose to use PD901 as opposed to the clinically relevant MEKi, trametinib. Some of the assays should be repeated with trametinib...the authors should perform Annexin V assays to determine the level of apoptosis using the MEKi + BETi combination".*

Response: We have now done this. As suggested by the reviewer, we have performed additional experiments combining the FDA-approved MEKi trametinib and the clinically relevant BETi OTX-

015. The results of those new experiments are completely consistent with the results of the experiments combining PD901 with BETi, therefore showing that the combination of either of two structurally different drugs targeting MEK and a BET inhibitor more potently induced cell death compared to single agents. These new data are now included in **new Appendix Fig. S4**, where cell death was assessed by staining cells with Annexin V and PI, and analyzed by flow cytometry, as in Appendix Fig S3E. We appreciate this suggestion, which has strengthened our conclusion.

We would also like to add that, although not FDA-approved, PD901 is currently being evaluated in several Phase I and Phase II clinical trials in combination with other anti-cancer agents, underscoring its clinical relevance (NCT 02022982, NCT 02096471, NCT02039336, NCT02510001).

Finally, we would like to mention that we had initially chosen PD-0325901 as we had an MTA for trametinib and we preferred to use a commercially available MEK inhibitor outside the MTA. Among the several well characterized MEK inhibitors currently available, we chose PD-901 as it has similar activity to trametinib *in vitro* and *in vivo* 1-3. We also have additional unpublished data showing almost identical activity of the non-ATP competitive allosteric inhibitors PD901 and trametinib inhibiting growth of NRAS mutant melanoma cells.

1. Wu PK, Park JI. MEK1/2 Inhibitors: Molecular Activity and Resistance Mechanisms. *Semin Oncol.* 2015; 42(6):849-62. \
2. Burgess MR, Hwang E, Firestone AJ, Huang T, Xu J, Zuber J, Bohin N, Wen T, Kogan SC, Haigis KM, Sampath D, Lowe S, Shannon K, Li Q. Preclinical efficacy of MEK inhibition in Nras-mutant AML. *Blood.* 2014; 124:3947-55.
3. Cheng Y, Tian H. Current Development Status of MEK Inhibitors. *Molecules.* 2017; 22. pii: E1551.

5. *“A model describing the findings should be incorporated into the main figures...”*

Response: We appreciate the suggestion and have done this. The model is now shown in **New Figure 7**. We have also included a visual abstract in the synopsis describing the findings and their efficacy in drug resistant tumors. The IPRES signature is now shown in expanded view **Fig EV4**.

Minor Comments:

1. *The introduction does not provide sufficient details and citations of the role of epigenetics in melanoma.*

Response: We regret that this was not clear; we have now expanded the introduction and added additional references about the role of epigenetics in melanoma (Pg. 3-4; highlighted text in the Introduction).

2. *“The Supp figures seem to be listed out of order in the manuscript and can be confusing to follow”.*

Response: We appreciate this recommendation and have now revised the order in which the supplementary figures are cited in the manuscript.

3. *“For TCF19 expression, the authors should also look at the genetic subgroups (Fig 3D is all TCGA patients) ...why does the KD of TCF19 induce apoptosis vs. cell cycle arrest? Is it Caspase-dependent?”*

Response: We have done this. Unfortunately, when dividing the data in genetic subgroups, the sample size is reduced, and the significance is markedly reduced. In the interest of rigor, we can only state that TCF19 levels are associated with survival of melanoma patients, and that there is no significant correlation with a particular genetic cohort.

With regard to KD of TCF19, we examined the effect of TCF19 silencing 9 days after transduction with TCF19 shRNA, as our goal was to determine if TCF19 could trigger cell death. Prompted by the reviewer’s question, we have examined the effects of TCF19 silencing on cell cycle progression at earlier time points.

We found that depletion of TCF19 three days post-transduction leads to an increase in the number of cells in G2/M (Appendix Fig S9). We now mention in the Results that both cell death and cell cycle arrest are occurring. We detect cleaved caspase 7 upon depletion of TCF19. However, treatment of cells transduced with TCF19 shRNA with the pan-caspase inhibitor Z-vad did not completely suppress cell death (data not shown). Overall, we feel we can only conservatively state that silencing TCF19 led to a combination of cell cycle arrest and cell death.

4. *LSL-NRAS model should be explained as a lox-stop-lox model. Were the injections of cells derived from the animals or cell lines? It is not clear.*

Response: We regret that this was not clear. Tumors were excised from LSL-NrasQ61R/Q61R (TpN16) mice and re-implanted into C57Bl6 syngeneic mice. The methods section describing the syngeneic tumor models has been revised to more clearly explain the models and tumors used (pg. 9: Methods/Animal models/ Syngeneic mouse models section).

5. *Pharmacological inhibition of BRD4 alone does not exist; it is inhibition of all BETs, see discussion.*

Response: We have revised the pertinent sentence to clearly indicate that pharmacological inhibition refers to the BET/BRD family of proteins (Pg.13). We thank the reviewer for pointing this out.

6. *Some of the Supp figures are missing statistics.*

Response: We have added statistics to all supplemental figures.

Referee #2

1. *Fig 1/S1 Define high and low expression of BET proteins (what percentile?)*

Response: We appreciate the suggestion; we have done this. We now define high and low expression of BET proteins in the methods section. For all associations of gene expression with survival, patients were split into two groups of high and low expression based on the gene's median expression level and tested using log-rank test. This figure has been modified following the recommendations of reviewer #1. Please see comment 1, Fig. 1A, EV Fig. 1, appendix Fig 1.

2. *Fig 3. The extent of cell death induced by knockdown of tcf19 is not consistent with the degree of knockdown, raising concerns that the effect of shRNA#11 are not specific. To be convincing, the authors should use an additional shRNA sequence or attempt to suppress the phenotype with tcf19 overexpression.*

Response: This is an excellent point. We have now optimized the conditions for TCF19 knockdown and used a different shRNA construct targeting the 3'UTR of TCF19. Similar to TCF19sh11, this hairpin triggers death of 70-80% of cells (See New **Figure 3D**). Additionally, as suggested by the reviewer, we have ectopically expressed TCF19 in cells transduced with TCF19 shRNA. Ectopic expression of TCF-19 suppresses the effects of TCF19 sh3'UTR thereby diminishing cell death induced by depletion of TCF19. These results support that the effects of shRNA-mediated TCF19 knockdown are specific.

Referee #3:

1. *"...Is there any correlation [between][BRD4 and TCF19 expression and NRAS mutation in TCGA data? Does BRD4 regulate the TCF19? Does the combination affect the BRAF mutant melanoma?"*

Response: These are excellent points. We have analyzed the TCGA data and found that:

- There is a moderate positive correlation between TCF19 and BRD4. BRD4 and TCF19 positively correlate in melanoma including NRAS and BRAF mutant tumors as well as NRAS/BRAF WT samples (New Appendix **Figure S8**).
- There is a correlation between BRD4 and NRAS, namely BRD4 is significantly higher in NRAS mutant tumors than in WT (P=0.03)
- There is a correlation between TCF19 and NRAS, namely TCF19 is significantly higher in NRAS

mutant tumors than in WT (P=0.008)

When we silenced BRD4, TCF19 is partially downregulated. This effect is greatly enhanced when we add MEKi (New Appendix **Fig S8A**). These results support that both BRD4 and the MAPK pathway are jointly required to regulate the expression of TCF19; we mention this in the text (pg X).

Further, we show in **Figure 4** that the combination is effective in a BRAF mutant syngeneic mouse model (BRA^{FV600E}/PTEN^{-/-}; Figure 4A) and in a PDX model derived from a BRAF/NRAS mutant melanoma patient who progressed on BRAFi therapy (Figure 4B). The combination of BETi/MEKi also effectively induces death of BRAF^{mut} and BRAF^{mut}/NRAS^{mut} melanoma short-term cultures derived from melanoma patients resistant to immunotherapy (BRA^{FV600E}: 13-456-3-3, 13-456-5-3; BRAF^{mut}/NRAS^{mut}: 15-1761-1-2, WM4231-2) (**Figure 4H**). Our data and previous studies from Dr. E. Hernando's group (Paoluzzi et al, 2016) indicate that BET inhibitors can potentiate MAPK inhibition (by either BRAFi or MEKi) in BRAF mutant melanoma.

2. “[On] Page 5, the authors state that ‘depletion of BRD4 decreased the viability of NRAS mutant melanoma cell, but induced only modest apoptosis, to have an explanation. Also, the degree of reduction of BRD4 does not correlate with apoptosis’”.

Response: We regret that this was not clear. The effect of depleting BRD4 was assessed by MTT assays (Figure 1D), which measures cell viability (combining cell death and proliferation arrest). In contrast, the data shown in Fig 1E was obtained using Annexin V/PI staining, which only quantifies dead (apoptotic and/or necrotic) cells. Both pieces of data support the premise that BRD4 silencing causes a decrease in cell viability without considerable cell death, suggesting that the main effect of BRD4 depletion is likely due to inhibition of cell proliferation, consistent with the data shown in Appendix Figure S3D-E. To further clarify this issue, we have now quantified the western blots for three independent experiments and are including a graph depicting the average expression of BRD4 in cells transduced with a vector control or BRD4shRNA (New Appendix Fig. S2B and primary data). This quantification depicts a more consistent correlation between BRD4 and BRD2 levels and cell death.

3. Figure 1 D lacks the label description of the box of light gray and dark gray.

Response: We thank the reviewer for the comment; we have moved the figure legend to make it more easily visible (Fig 1)

4. In page 5, the authors state that [a] ‘single dose of either compound as a single agent transiently restrained cell proliferation, but this effect was not sustained at 14 days’, to explain

Response: We regret that this was not clear. In the experiments shown in Appendix Fig. S3D, we treated NRAS mutant melanoma cells and human primary fibroblasts with a single dose of JQ1 (0.5 μM), PD901 (0.1 μM) or the combination of both drugs at the same dose. Cells were then collected, stained and analyzed after 7 or 14 days of treatment. In these experiments, we found that after 7 days of treatment with a single dose, all drug treatments similarly prevented the growth of NRAS mutant melanoma cells. However, when cells were analyzed after 14 days of treatment, the effect of single agent BETi or MEKi was lost; the cells recovered and continue to proliferate. In contrast, the cells that were treated with the drug combination remained arrested and were not able to regrow. We thank this Reviewer for pointing this out, and we now make this point more clearly in the Results section.

5. “In page 8-9, the authors state that ‘depletion of TCF19 let to activation of caspase-7...the degree of reduction of TCF19 seems not correlate with caspase-7...’”

Response: Optimized conditions and new shRNAs have been used to repeat these experiments. Please see response to reviewer #2 comment 2 and new data in Fig. 3C-D

6. “The sample size (cohort) in Figure 2 C to E is small (n=3)”

Response: We appreciate the reviewer's comment. As an explanation, the data shown in figure 2C was from a pilot study, which was performed to assess feasibility, in order to estimate sample size for follow-up studies. This pilot study was performed using a tool compound, JQ1. Following these

promising data we opted to perform a larger study with the more clinically relevant compound, OTX-015. Taken together, the data from both of these compounds is corroborative. We hope this Reviewer understands our reasoning for this smaller initial cohort.

7. *“In Figure G, the level of TCF19 expression should [be] show[n] (such as by PCR or IHC)”*.

Response: We have done this. We now include in Appendix Table S5 the mRNA expression values for TCF19 to illustrate the changes in TCF-19 RNA levels at baseline, post-treatment or progression for responders and non-responders. The charts shown in Figure 6G were generated from RNA-sequencing data from patient samples. We have now added data from three additional patient samples that became available after our initial submission. We hope this addresses this Reviewer’s concern. In sum we hope the Reviewers feel that we have addressed their critiques clearly and completely, and that they can now offer improved enthusiasm and acceptance of this work.

2nd Editorial Decision

22 February 2018

Thank you for the submission of your revised manuscript to EMBO Molecular Medicine. We have now received the enclosed reports from the referees that were asked to re-assess it. As you will see the reviewers are now globally supportive and I am pleased to inform you that we will be able to accept your manuscript pending final editorial amendments.

Please submit your revised manuscript within two weeks.

I look forward to reading a new revised version of your manuscript as soon as possible.

2nd Revision - authors' response

07 March 2018

We have revised our manuscript per the journal requirements.

Corresponding Author Name: Jessie Villanueva
 Journal Submitted to: EMBO Molecular Medicine
 Manuscript Number: EMM-2017-08446